

# Soil carbon, nitrogen, and phosphorus storage in juniper-oak savanna: Role of vegetation and geology

Che-Jen Hsiao[1,2], Pedro A. M. Leite[1], Ayumi Hyodo[1], Thomas W. Boutton[1]

[1]Department of Ecology and Conservation Biology, Texas A&M University, College Station, Texas 77843, USA
[2]Present address: Department of Soil, Water, and Climate, University of Minnesota, St, Paul, Minnesota 55108, USA

*Correspondence to*: Che-Jen Hsiao (chsiao@umn.edu)

**Abstract.** Woody plant encroachment into grasslands and savannas has been globally widespread during the past century, likely driven by interactions between grazing, fire suppression, rising atmospheric $CO_2$, and climate change. In the southernmost U.S. Great Plains, Ashe juniper and live oak have increased in abundance. To evaluate potential interactions

between this vegetation change and the underlying soil parent material on ecosystem biogeochemistry, we quantified soil organic carbon (SOC), total nitrogen (TN), total phosphorus (TP), and $\delta^{13}C$ of SOC in soils obtained from trenches passing through grassland, juniper, and oak patches on soils lying atop Edwards vs. Buda limestone formations in central Texas. Soils on the Edwards formation are more shallow and have more rock outcropping than those on Buda. The $\delta^{13}C$ of SOC under grasslands was -19 ‰, while those under woody patches were -21 to -24 ‰, indicating wooded areas were relatively recent

components of the landscape. Compared to grasslands, areas now dominated by juniper or oak had elevated SOC, TN, and TP storage in soils lying atop Edwards limestone. In Buda soils, only oak patches had increased SOC, TN, and TP storage compared to grasslands. Woody encroachment effects on soil nutrients were higher in soils on the Edwards formation, perhaps because root and litter inputs were more concentrated in the relatively shallow layer of soil atop the Edwards bedrock. Our findings suggest geological factors should be considered in predicting responses of nutrient stores in savannas following

vegetation change. Given that woody encroachment is occurring globally, our results have important implications for the management and conservation of these ecosystems. The potential interactive effects between vegetation change and soil parent material on C, N, and P storage warrant attention in future studies aimed at understanding and modeling the global consequences of woody encroachment.

**Plain Language Summary.** During the past century, tree cover has increased in grasslands, savannas, and other dryland ecosystems around the world due to grazing, fire suppression, rising atmospheric $CO_2$, climate changes, and their interactions. In grasslands of the southern Great Plains USA, juniper and oak trees have increased in abundance. We examined how this vegetation change interacts with differences between soils lying atop different geological formations to influence soil carbon (C), nitrogen (N), and phosphorous (P) storage in soils. Concentrations of soil C, N, and P were significantly higher under oak

and juniper trees than under grasslands on soils derived from both the Edwards and Buda geological formations; however, those elements increased more under oak and juniper growing on Edwards soils than on Buda soils. We speculate that higher



soil C, N, and P concentrations in soils atop the Edwards limestone may be due to the shallow depth to bedrock (40 cm) that constrains root and litter inputs to a more limited soil volume. Given the global extent of woody encroachment, we suggest that interactions between vegetation change and geology warrant consideration in future studies, and could improve efforts to
predict and model soil C, N, and P storage in grasslands, savannas, and other dryland ecosystems.

**1 Introduction**

        Woody plant encroachment into grasslands, savannas, deserts, and other semiarid and arid ecosystems is a globally extensive land cover change that has been occurring during the past 150 yrs (Archer et al., 2017; Sala and Maestre, 2014; Stevens et al., 2017). This important vegetation change is driven by several potentially interacting local and global phenomena,
including elimination of naturally occurring fires, chronic livestock grazing, rising atmospheric $CO_2$ concentrations, and climate change (Archer et al., 2017; Rosan et al., 2019; Stevens et al., 2017; Venter et al., 2018; Zhou et al., 2019). Rates of increase in woody cover range from 0.1 to 2.3 % $yr^{-1}$ in grasslands and savannas throughout the world (Archer et al., 2017; Deng et al., 2021). These changes in the relative abundances of woody plants and grasses often have significant ecological impacts on arid and semi-arid ecosystems. For example, increases in above- and belowground inputs of organic matter derived
from woody plants often alter soil carbon (C), nitrogen (N), and phosphorus (P) storage, cycling rates, and stoichiometric relationships (Barger et al., 2011; Eldridge et al., 2017; Finzi et al., 2011; Zhou et al., 2018a).

        Little is known regarding how geology might interact with changes in woody plant cover to influence ecosystem biogeochemistry. Geological factors play key roles in determining fundamental soil properties, including mineralogy, chemistry, and texture, which in turn can be important determinants of soil C, N, and P biogeochemistry (Augusto et al., 2017;
Farella et al., 2020). The reactivity of the soil mineral phase can change the stabilization potential of soil organic carbon (SOC) under similar vegetation (Doetterl et al., 2015; Torn et al., 1997), and clayey soils have greater potential to retain mineral-derived nutrients and organic matter across a broad range of soil substrates (Soong et al., 2020). Furthermore, the presence of living organisms (plants and microbes) in the weathering zone of geological materials can influence soil formation and properties through (i) mechanical processes, (ii) changes in soil pH, and (iii) root and microbial exudates that influence
hydrolysis reactions and chelation (Anderson, 1988). Since plant species can differ greatly in root distribution patterns, root strength, and root exudate production (Canadell et al., 1996; Dietz et al., 2020; O'Keefe et al., 2021), their interactions with parent material may be different, thus having important influences on soil development and biogeochemistry.

        Throughout the Great Plains of North America, *Juniperus* spp. have increased their abundance during the past century in areas that were previously grass-dominated (Leis et al., 2017; Smeins and Merrill, 1988; Van Auken and Smeins, 2008).
Rates of *Juniperus* encroachment reported for this region between 1990-2010 ranged from approximately 30-40 $km^2$ $yr^{-1}$ in Kansas and Oklahoma to 85 $km^2$ $yr^{-1}$ in Nebraska (Meneguzzo and Liknes, 2015; Wang et al., 2018a). In Texas, *Juniperus* cover increased at a rate of 441 $km^2$ $yr^{-1}$ between 1948-1982 (Ansley and Wiedemann, 2008). These dramatic changes in *Juniperus* cover have been shown to alter key ecosystem properties such as primary productivity, evapotranspiration rates, soil



water infiltration and percolation depth, and pool sizes and flux rates of essential elements (Jessup et al., 2003; Leite et al.,
2020; McKinley and Blair, 2008; Wang et al., 2018b; Zhang et al., 2023).

In the southernmost portion of the North American Great Plains, the Edwards Plateau spans approximately 100,000
km² of central Texas and is covered with savannas and woodlands dominated primarily by the woody species *Juniperus ashei
Buchholz* (Ashe juniper) and *Quercus virginiana Mill.* (live oak), as well as grasses and forbs typical of the southern mixed-
grass prairie (Smeins et al., 1976; Smeins and Merrill, 1988; Taylor et al., 2012). Although juniper and oak have been long-
term woody components of this landscape, juniper populations in particular have increased markedly during the past century,
largely due to reduced fire frequencies caused by chronic livestock grazing and fire suppression policies (Ansley and
Wiedemann, 2008; Bendevis et al., 2010; Leite et al., 2020; Wilcox et al., 2007; Marshall, 1995). Woody plant cover in this
region increased from 12 to 40 % between 1949 to 1990, with over half of this increase attributable to Ashe juniper (Fuhlendorf
et al., 1996). Despite these dramatic vegetation changes, little is known how woody encroachment in the Edwards Plateau
region might influence soil storage of C, N, P and other important elements. In addition, soils of the Edwards Plateau are
derived from multiple geological formations, and it remains unknown how woody encroachment might alter nutrient stores in
soils derived from different parent materials. At the western edge of the Edwards Plateau, most soils lie atop the Buda and
Edwards geological formations. Soils on the Edwards formation are shallow with extensive areas of rock outcropping, while
those on the Buda limestone are generally deeper and contain soft marl layers on top of the hard limestone (Gabriel et al.,
2009; Wilcox et al., 2007).

The purpose of this study was to evaluate the long-term impacts of woody plant encroachment on the biogeochemistry
of soil C, N, and P throughout soil profiles lying on top of Edwards vs. Buda limestone in juniper-oak savannas of the Edwards
Plateau in the southern Great Plains. To accomplish this, soil samples were collected to the depth of bedrock from multiple
trenches that passed through both grassland and woody plant communities. We used elemental analyses to quantify pool sizes
of SOC, total N (TN), and total P (TP) in soils, and measured δ¹³C values of those same soils to verify woody encroachment
and to estimate the relative proportions of grass vs. woody sources of soil organic matter. We tested the hypotheses that: (1)
woody plants are relatively recent components of grasslands on the western Edwards Plateau; (2) soils under woody plants
will have larger pool sizes of SOC, TN, and TP than grass-dominated areas; and (3) vegetation cover will interact with
geological substrate to influence SOC, TN, TP, and soil inorganic carbon (SIC).

## 2 Materials and Methods

### 2.1 Site description

Research was conducted at the Texas A&M AgriLife Sonora Research Station on the western edge of the Edwards
Plateau, Texas (30°15' N, 100°33' W; altitude 670 m). The mean annual temperature was 19.3 °C and the mean annual
precipitation was 476 mm from 2012 to 2020 (Cooperative Climatological Data Summaries, 2022). Summers are hot and dry
with average July temperature of 30 °C, while winters are mild with average January temperature of 10 °C. Most precipitation



occurs during May-June and September-October, and is highly variable between and within years. The Koppen-Geiger climate classification for this region is hot arid steppe (Beck et al., 2018).

The Edwards Plateau is an uplifted and dissected limestone plateau (karst topography) with gentle slopes. Soils in this region are clayey Mollisols with shallow soils on plateaus and hills, and deeper soils on plains and valley floors (Gabriel

et al., 2009; Wiedenfeld and McAndrew, 1968). The predominant soil types on shallow soils on plateaus and hills are Eckrant and Tarrant soil series (clayey-skeletal, smectitic, thermic Lithic Calciustolls) which lie atop the Edwards formation (Wilcox et al., 2007). These soils contain large amounts of limestone fragments and limestone outcrops. Depth to bedrock for these soils was generally < 0.4 m (Figs. 1 and S1). The clay content in the top 5 cm of Edwards soil is 30-40% and increases with depth to almost 50% at 20 cm depth (Marshall, 1995). Soils occurring on plains and valley floors are Prade and Valerna soil

series (Loamy, mixed, superactive, thermic shallow Petrocalcic Calciustolls) which lie atop the Buda formation (Wilcox et al., 2007). These soils are generally deeper than those lying atop the Edwards formation and contain hard limestone and soft marl layers on top of the limestone bedrock (Figs. 1 and S1). The marl layer (Bkkm horizon) is comprised of weathered carbonate and cements the B horizon with soil particles (Soil Survey Staff, 2014). The clay content in Buda soil is generally 2-5 % lower than that in Edwards soil based on USDA/NRCS data (Official Soil Series Descriptions (Eckrant series), 2022; Official Soil

Series Descriptions (Rio Diablo series), 2022; Official Soil Series Descriptions (Ector series), 2022; Official Soil Series Descriptions (Valera series), 2022; Official Soil Series Descriptions (Prade series), 2022). Depth to consolidated bedrock for Buda soils was approximately 2 m. Although the hard limestone geological formations underlying both the Edwards and Buda soils have contributed somewhat to the formation of these soils, there is considerable chemical and physical evidence indicating that these soils are derived largely from an overlying limestone residuum with distinctly different attributes than the underlying

limestone (Rabenhorst and Wilding, 1986a, b; Cooke et al., 2007). The Del Rio Clay, an Upper Cretaceous marly limestone that locally overlies the Edwards limestone, has been proposed as the dominant source of these soils (based on texture, mineralogy, and Nd isotope composition), at least on the eastern portion of the Edwards Plateau (Cooke et al., 2007).





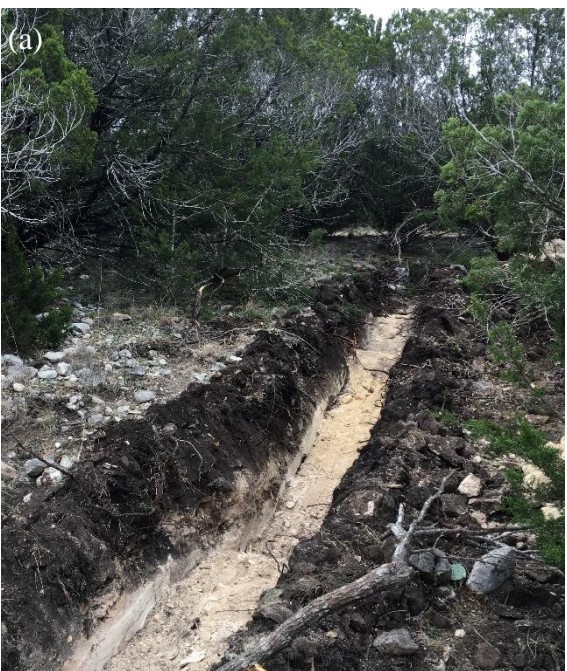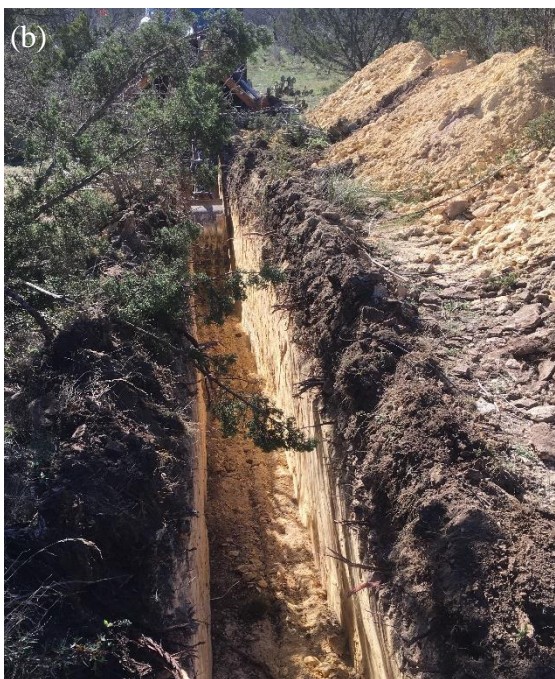

**Figure 1.** Trench 1 in Edwards soil (a) and Trench 5A in Buda soil (b) at Texas A&M AgriLife Sonora Research Station on the Edwards Plateau, Texas. The beige color in (a) is limestone. The yellow-whitish layer in (b) is marl soil in which the individual soil particles are aggregated with calcium carbonate.

The study area lies at the southernmost extent of the Great Plains, and the contemporary vegetation of the site is typical of this region. Dominant woody species are live oak and Ashe juniper. Other less abundant woody species include *Quercus pungens Liebmann* (scrub oak), *Juniperus pinchotii Sudw* (redberry juniper), and *Prosopis glandulosa* Torr. (honey mesquite). Woody species occur commonly in small clusters or as isolated individuals within the surrounding grassland matrix. Dominant grass species include *Hilaria belangeri (Steud) Nash* (curly mesquite), *Bouteloua curtipendula (Michx.) Torr.* (sideoats grama), *Aristida* spp. (three awns), and *Nassella leucotricha (Trin & Rupr.) Pohl* (Texas wintergrass). Grazing in this area was heavy to moderate from approximately 1880 to 2010, but the area chosen for this study was grazed lightly and intermittently for the past 10 yrs (Leite et al., 2020).

**2.2 Field sampling**

Soils were sampled from three trenches lying atop the Edwards and three lying atop the Buda formation using a backhoe in February 2019. Trench locations were selected to represent (i) the major geological formations underlying the soils in the study area (Edwards and Buda limestone) based on the USDA/NRCS Web Soil Survey (Web Soil Survey, 2022), and (ii) the major vegetation types, including live oak, Ashe juniper, and herbaceous species occurring on the area. Trench lengths and depths were variable, depending on the depth to bedrock and the distribution of bare rock at the surface (Table 1). At 2 m



intervals along each trench face, soil samples were collected from multiple depth intervals (0-10, 10-20, 20-40, 40-70, 70-100, 100-120, 120-150 cm), although depth to bedrock sometimes prevented acquisition of the deeper depths in some trenches, particularly those atop the Edwards formation. Within the middle of each depth increment, two soil cores (7.6 cm width x 10 cm length) were inserted horizontally into the trench face. One core was used for determination of soil bulk density and root

biomass, while the other was used for pH, elemental concentrations, and stable isotope analyses. Aboveground vegetation cover within 2 m of each sampling location was mapped and categorized as grassland, Ashe juniper, live oak, or mesquite based on dominant vegetation types. In order to characterize the isotopic composition of plant inputs to the soil organic matter pool, approximate equal amounts of live leaves from three individuals of Ashe juniper, live oak, and the dominant grass species were collected near each trench. Litter from Ashe juniper, live oak, and dominant grass species together with standing dead

grass materials were collected from 0.25 m × 0.25 m plots with three replications close to each leaf sampling location.

**Table 1.** Geological substrates and physical dimensions of soil trenches

| Trench | Geological formation | Average depth (m) | Length (m) |
|--------|---------------------|-------------------|------------|
| 1 | Edwards | 0.7 | 30 |
| 2 | Edwards | 0.4 | 8 |
| 4 | Edwards | 0.4 | 14 |
| 5A | Buda | 1.5 | 20 |
| 5B | Buda | 1.4 | 9 |
| 6 | Buda | 1.4 | 12 |

### 2.3 Laboratory analyses

One soil core from each sampling point was oven-dried at 105 °C overnight, and then passed through a 2 mm sieve to remove rock fragments, and to retrieve fine roots for quantification of root densities. Soil bulk density was measured and reported on a gravel-free basis (Culley, 1993). The other soil core was air-dried, passed through a 2 mm sieve to remove rocks and large organic fragments, and subsampled for soil pH using 1:2 soil/0.01M $CaCl_2$ slurry. Another subsample of the < 2 mm fraction was pulverized in a centrifugal mill (Angstrom Inc., Belleville, MI, USA) and used to quantify concentrations of SOC,

SIC, TN, and TP, as well as the $\delta^{13}C$ of SOC. Leaf and litter samples were oven-dried at 80 °C for 48 h and pulverized in a centrifugal mill.

Pulverized leaf tissues, litter, and soils were analyzed for C and N concentrations and $\delta^{13}C$ values by dry combustion using a Costech ECS 4010 elemental analyzer (Costech Analytical Technologies Inc., Valencia, CA, USA) interfaced via a ConFlo IV with a Delta V Advantage isotope ratio mass spectrometer (Thermo Scientific, Bremen, Germany) in the Stable

Isotopes for Biosphere Sciences Laboratory, Texas A&M University. Pulverized subsamples were weighed into tin capsules to measure the percentage of total C and total N. For soils, a second subsample was weighed into a silver capsule and treated with 3N HCl to remove $CaCO_3$ until no reaction occurred (Nieuwenhuize et al., 1994). This subsample was used to measure the concentration and $\delta^{13}C$ value of SOC. Soil inorganic carbon concentrations were computed as the difference between the





measurement of total C derived from the non-acid treated soil and organic C derived from the acid treated soil. Stable C

isotopic ratios were presented in δ notation according to Eq. 1:

$$\delta = \left[ \frac{R_{\text{sample}} - R_{\text{STD}}}{R_{\text{STD}}} \right] \times 10^3 \tag{1}$$

where $R_{\text{sample}}$ and $R_{\text{STD}}$ are the $^{13}\text{C}/^{12}\text{C}$ ratio of the sample and standard, respectively. The $^{13}\text{C}/^{12}\text{C}$ ratios were expressed relative to the Vienna PeeDee Belemnite (V-PDB) standard (Coplen, 1994). Precision of duplicate measurements was 0.1 ‰ for $\delta^{13}\text{C}$. Soil TP was measured using sulfuric acid digestion and colorimetry (Dick and Tabatabai, 1977) by the Soil Testing Laboratory at Kansas State University, Manhattan, KS.

The fraction of soil C derived from $C_4$ grasses was calculated by the mass balance equation Eq. 2:

$$\delta^{13}C_{\text{soil}} = \delta^{13}C_{\text{grass}}(f) + \delta^{13}C_{\text{woody tissue}}(1 - f) \tag{2}$$

where $\delta^{13}C_{\text{soil}}$, $\delta^{13}C_{\text{grass}}$, and $\delta^{13}C_{\text{woody tissue}}$ were the $\delta^{13}\text{C}$ values of soil, $C_4$ grass, and leaf tissues from $C_3$ woody plants (live oak or Ashe juniper), respectively. The fraction f was the proportion of SOC derived from $C_4$ plants and $(1 - f)$ is the proportion of SOC derived from $C_3$ plants.

## 2.4 Calculation of soil C, N, and P densities

Since woody encroachment often changes soil bulk density (Hudak et al., 2003; Throop et al., 2012; Throop and Archer, 2008), quantification of SOC, TN, TP, and SIC densities in different soil layers were performed on the basis of equivalent soil mass (Fowler et al., 2023). In addition, we limited our statistical analyses of soil variables to the 0-10, 10-20, and 20-40 cm depth increments to facilitate comparisons due to differences in maximum soil depth between trenches. The estimation of SOC stocks (Mg C ha$^{-1}$) was calculated considering SOC concentrations and soil masses in the 0-10, 10-20, and

20-40 cm soil layers along with the trench faces by cubic spline function (Fuentes et al., 2010; von Haden et al., 2020; Wendt and Hauser, 2013). Although all SOC stocks were calculated and compared as equivalent soil masses, the results were expressed as 0-10, 10-20, and 20-40 cm soil layers for better clarity. To facilitate comparisons between the unequal sampling depth intervals, we conducted our analyses on SOC density (kg C m$^{-3}$), computed using the following Eq. 3:

$$SOC\ density\ (kg\ C\ m^{-3}) = \frac{SOC\ stock\ (Mg\ C\ ha^{-1})}{depth\ interval\ (cm)} \times 10 \tag{3}$$

Total N, TP, and SIC density were calculated and presented using the same process.

## 2.5 Data analyses

The effects of geology (Edwards or Buda formations), vegetation type (grassland, Ashe juniper, live oak, or mesquite), and soil depth on soil chemical and isotopic measurements were analyzed using a three-way analysis of variance (ANOVA) with repeated measures in a linear mixed-effects model with restricted maximum likelihood estimations. Trench ID was treated as a random factor. Sampling locations along the trench face and depth were treated as a random factor with repeat

measurements. Post hoc multiple comparisons were made using the Tukey adjustment. Dependent variables were tested for



the assumption of normality using the Shapiro-Wilk test and then logarithmic or square-root transformed when necessary to meet assumptions of normality before analysis. The significance of ANOVA outputs was evaluated using an α value of 0.05. All errors were reported as standard errors. All statistical analyses were performed in R (R Core Team, 2013), including package nlme (Pinheiro et al., 2012) for ANOVA. To visualize soil properties in horizontal and vertical dimensions along the trench faces, contour maps were developed utilizing all data from all soil depths. Contour maps were created using the local weighted regression surface (LOWESS) function in R.

## 3 Results

### 3.1 Soil characteristics

Soil bulk densities were significantly affected by the depth x geology interaction (Table 2), increasing more rapidly with depth in Edwards vs. Buda soils (Fig. 2a). Soil bulk densities were significantly higher beneath grasslands than those under live oak or Ashe juniper. The extremely high abundance of gravel and rock fragments in Edwards soils under live oak and Ashe juniper canopies prevented measurements of bulk density at depths > 20 cm. Soil pH was significantly affected by both soil depth and vegetation type (Table 2), increasing with depth, and lower beneath oak than beneath juniper and grass throughout the 0-40 cm profile (Fig. 2b). Fine root densities were affected significantly by the depth x vegetation interaction (Table 2), with higher densities in soils beneath oak and juniper canopies than in grasslands soils below the 10 cm depth (Fig. 2c). Soil inorganic C density expressed based on equivalent soil mass was affected by a complex depth x vegetation x geology interaction (Table 2). This interaction was attributable to greater differences between vegetation types on Edwards than Buda soils, and to the fact that SIC increased more strongly with soil depth beneath oak than beneath grassland or juniper vegetation (Fig. 2d). Lower SIC values beneath oak and juniper on the Edwards soil were related to lower soil pH values beneath their canopies; similarly, lower soil SIC values beneath oak on the Buda soils were also related to lower soil pH.

**Table 2.** Results from ANOVA testing for effects of soil depth, vegetation, geology, and their interactions on soil attributes. Abbreviations are SOC, soil organic carbon; TN, total nitrogen; TP, total phosphorous; C:N, C:N ratio; C:P: C:P ratios; N:P, N:P ratios; BD, bulk density; SIC, soil inorganic carbon; fine roots, fine root (< 2 mm) density. Asterisks label significant changes: $p < 0.05$; $p < 0.01$; $p < 0.001$; ns, non-significant.

| | SOC | TN | TP | SIC | $\delta^{13}$C | C:N | C:P | N:P | BD | pH | Fine roots |
| | (kg m⁻³ soil) | | | | (‰) | | | | (g cm⁻³) | | (g m⁻³) |
|---|---|---|---|---|---|---|---|---|---|---|---|
| depth (D) | ns | ns | ns | *** | *** | ns | *** | *** | *** | ** | ns |
| vegetation (V) | *** | *** | * | *** | *** | ns | *** | *** | ** | *** | ns |
| geology (G) | *** | *** | *** | ns | ns | ns | * | * | ns | ns | ns |
| D × V | ns | ns | * | ** | ns | ns | ns | ns | ns | ns | * |
| D × G | * | * | * | ns | ns | ns | ns | ns | * | ns | ns |
| V × G | *** | *** | ** | *** | ns | ** | * | ns | ns | ns | ns |
| D × V × G | ns | ns | ns | * | ns | ns | * | ns | ns | ns | ns |





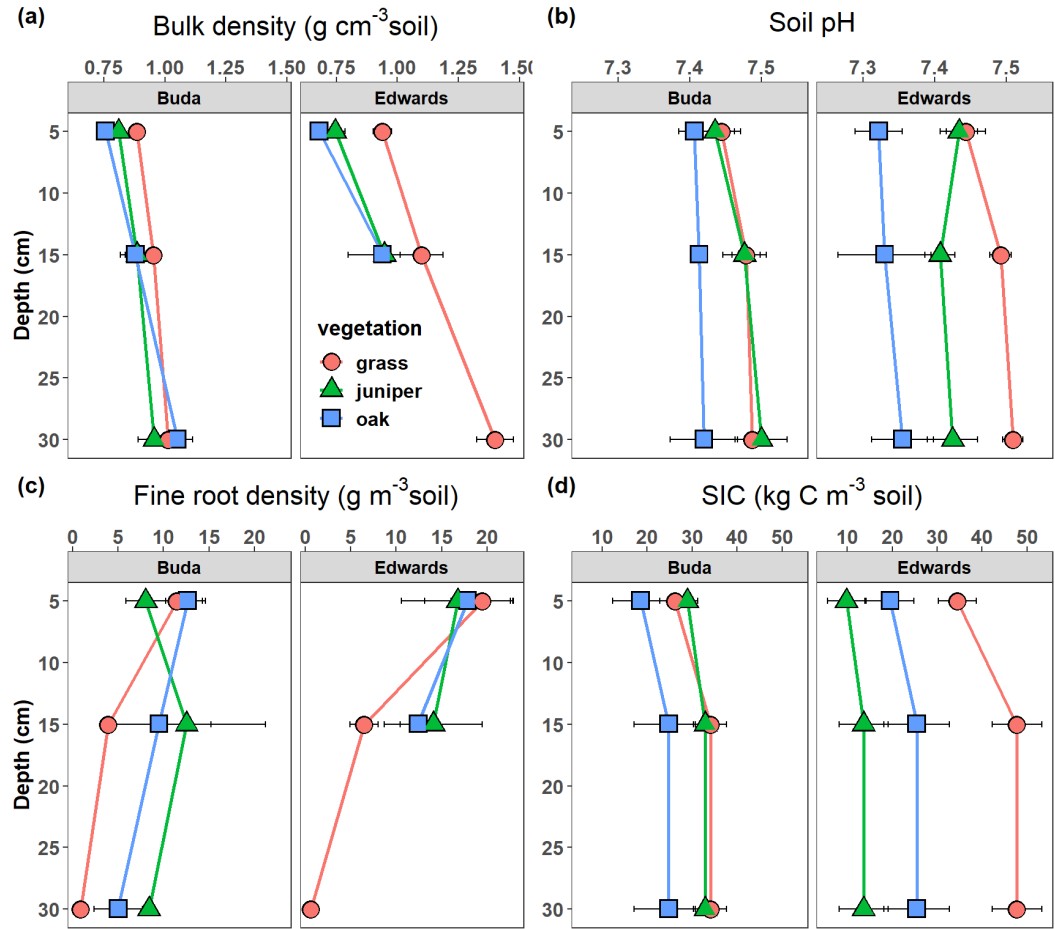

**Figure 2.** Vertical changes in (a) soil bulk density, (b) pH, (c) fine root biomass density and (d) soil inorganic carbon (SIC) density beneath grass, juniper, and oak in soils derived from Buda vs. Edwards formations. The extremely rocky nature of the soil below 20 cm depth beneath oak and juniper canopies precluded measurements of bulk density and root density in the Edwards trenches. Results are given as means ± standard errors of the mean. Data are plotted at the midpoints of the depth increments.

### 3.2 δ¹³C values of soils and litter

Litter $\delta^{13}C$ values were -26.7, -24.7, and -17.1 ‰ under live oak, Ashe juniper, and grass, respectively (Fig. 3). The $\delta^{13}C$ of SOC was significantly affected by vegetation and soil depth (Table 2). The $\delta^{13}C$ values of SOC under grasslands were -19 and -20 ‰ in the 0-10 cm layers in Edwards and Buda soils, respectively (Fig. 3a). Soil $\delta^{13}C$ values under woody patches were lower than grasslands on both Edwards and Buda soils, ranging from approximately -24 to -20 ‰. The $\delta^{13}C$ of SOC increased with depth beneath all vegetation types and on both Buda and Edwards soils. The $\delta^{13}C$ values gradually increased from the centers of woody patches to their edges, and reached highest values in the grasslands (Figs. S1 and S2).




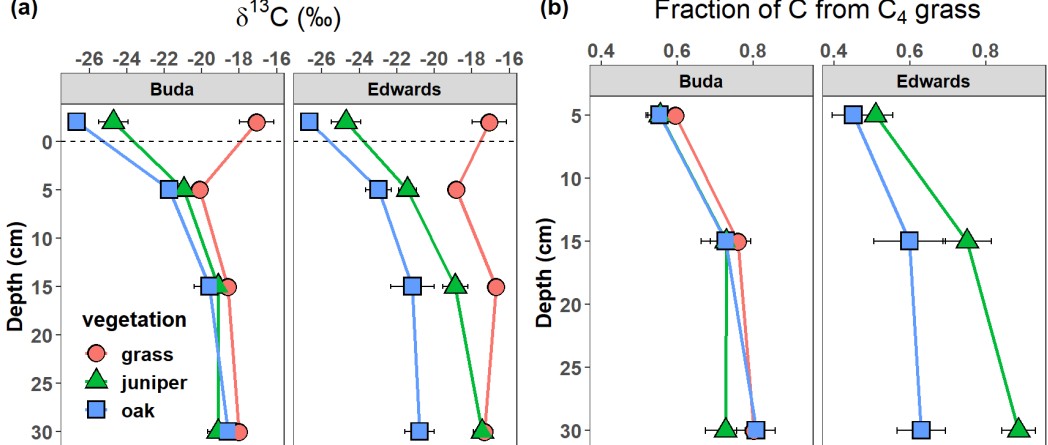

**Figure 3.** Changes in (a) litter and soil $\delta^{13}C$, and (b) the fraction of SOC derived from $C_4$ grass beneath juniper and oak in soils derived from the Buda vs. Edwards formations calculated using mass balance. Results are given as means ± standard errors of the mean. Data are plotted at the midpoints of the depth increments.


The average $\delta^{13}C$ value for $C_4$ grasses was -16.3 ± 0.8 ‰, while that for $C_3$ woody plants was -28.5 ± 0.5 ‰ for live oak and -26.8 ± 0.4 ‰ for Ashe juniper. These values were applied to a mass balance equation (Eq. 2) to estimate the relative proportion of SOC originating from $C_4$ grasses. The fraction of SOC from $C_4$ grass was approximately 0.5 in the 0-10 cm depth and increased up to 0.7-0.8 in the 20-40 cm depth increment (Fig. 3b). Soils under oak had significantly less C derived from

$C_4$ grass than juniper on the Edwards soils.

**3.3 Soil organic C, TN, TP, and their stoichiometric ratios**

The SOC, TN, and TP densities (kg m$^{-3}$ soil on an equivalent soil mass basis) in the upper 40 cm of the profile were all significantly affected by the vegetation x geology interaction (Table 2). Soils beneath live oak and Ashe juniper had higher SOC and TN than grasslands throughout the profile on Edwards soils, while oak had higher SOC and TN on Buda soils (Figs.

4a and c). The cumulative SOC in the 0-40 cm profile on Edwards soils was 23.1 and 21.0 kg C m$^{-2}$ in soils under juniper and oak, respectively, compared to 15.5 kg C m$^{-2}$ for the grassland (Fig. 4b). In contrast, cumulative SOC on Buda soils was significantly lower, with 14.2, 10.4, and 10.9 kg C m$^{-2}$ in soils under oak, juniper, and grassland, respectively. The vertical profiles of TN were similar to SOC (Fig. 4c). Cumulative TN (0-40 cm) was greater in Edwards (1.35-1.85 kg N m$^{-2}$) than Buda soils (0.85-1.12 kg N m$^{-2}$), and greater in oak than in juniper or grassland on Edwards soils (Fig. 4d). On Buda soils, oak

was significantly higher in cumulative TN (1.1 kg N m$^{-2}$) than grassland (0.8 kg N m$^{-2}$). Soils under oak had higher TP densities (kg P m$^{-3}$ soil) than either grassland or juniper throughout the 0-40 cm profile on Edwards soils (Fig. 4e). The cumulative TP in the 0-40 cm profile on Edwards soils was highest beneath oak (0.254 kg P m$^{-2}$), followed by grass (0.210 kg P m$^{-2}$) and juniper (0.165 kg P m$^{-2}$) (Fig. 4f). Buda soils were generally lower in TP than Edwards soils, and there was no significant difference in cumulative TP between the three vegetation types in Buda soils.





**Figure 4.** Vertical changes in (a) soil organic carbon (SOC), (c) total nitrogen (TN), and (e) total phosphorus (TP) density (kg m$^{-3}$) beneath grass, juniper, and oak in soils derived from Buda vs. Edwards formations. Data are plotted at the midpoints of the depth increments. Cumulative SOC (b), TN (d), and TP (f) in the full 0-40 cm profile are shown with significant differences indicated by asterisks: *p < 0.05,

\*\*p < 0.01, \*\*\*p < 0.001; ns, non-significant. Results are given as mean ± standard error. Different letters represent significant differences between vegetation types (p < 0.05) within specific geology formation.

Soil organic C concentrations (g C kg$^{-1}$ soil) were affected by the depth x vegetation x geology interaction, while TN concentrations were affected by geology and the depth x vegetation interactions (Table S1). For both SOC and TN concentrations, soils under grasslands were the lowest among three vegetation types on both Buda and Edwards soils (Fig. S3). In addition, SOC and TN concentrations were significantly higher on Edwards than on Buda soils. Soil TP concentration was affected by the vegetation x geology interaction (Table S1), with oak on Edwards soils having the highest TP compared to other vegetation types on either Edwards or Buda soils (Fig. S3).

Soil C:N ratios were significantly affected by the vegetation x geology interaction (Table 2), with higher ratios beneath oak and juniper than in grassland on Edwards soils throughout the 0-40 cm soil profile (Fig. 5a). However, on Buda soils, the C:N ratio was greater in oak and grass than juniper throughout the profile. Soil C:P ratios were significantly affected by the vegetation x geology x depth interaction due to: (i) larger differences between grasses vs. woody plants in the Edwards than in the Buda soils, (ii) lower values on the Buda than the Edwards soils, and (iii) differences in the depth responses between the plant species (Fig. 5b). Soil N:P ratios were significantly affected by the direct effects of vegetation, geology, and depth, but not influenced by any interactions (Table 2). The N:P ratios were generally higher beneath woody plants than grasslands, higher in the Edwards than Buda soils, and decreased with soil depth.



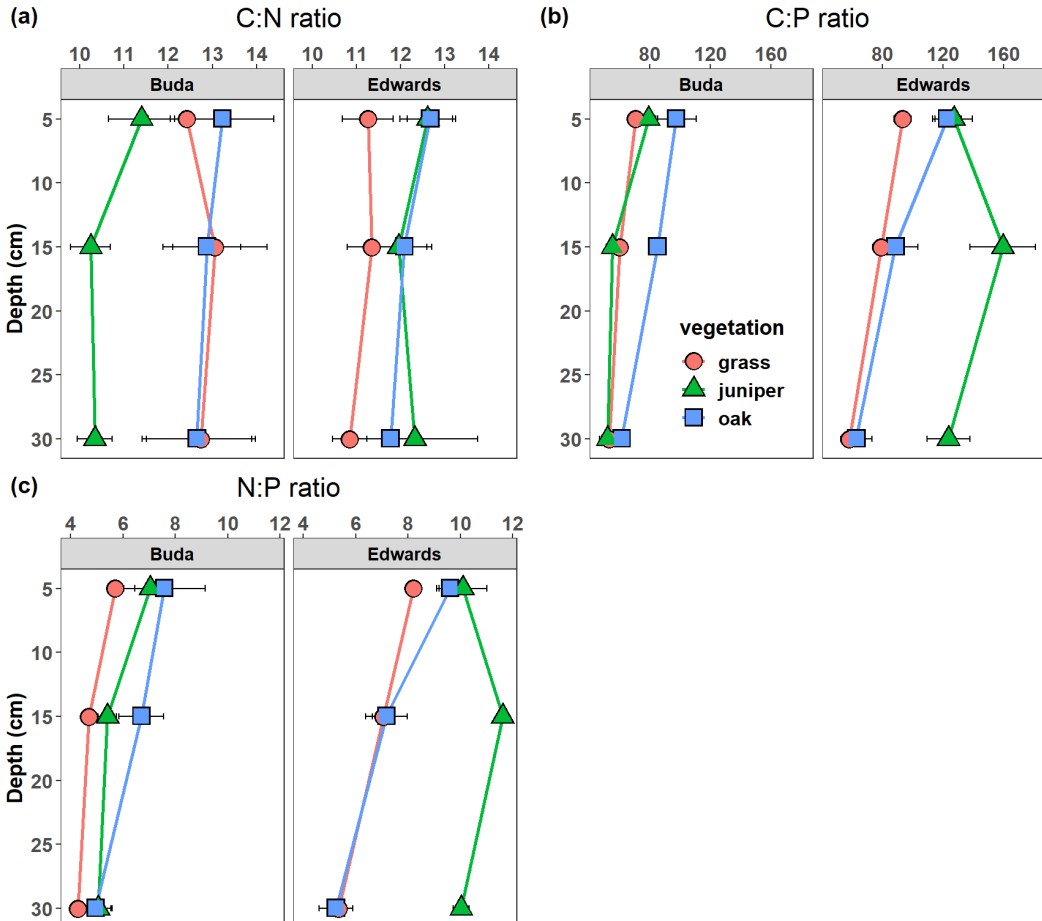

**Figure 5.** Changes in (a) soil C:N ratio, (b) soil C:P ratio, and (c) soil N:P ratio with soil depth beneath grass, juniper, and oak canopies on Buda vs. Edwards formations. Results are given as means ± standard errors of the mean. Data are plotted at the midpoints of the depth increments.

## 4 Discussion

### 4.1 Evidence for vegetation change

The $\delta^{13}C$ values of surface soils (0-10 cm) beneath juniper (-21 ‰) and oak (-23 ‰) canopies are higher than those obtained for litter (juniper = -25 ‰; oak = -27 ‰) and fresh leaf tissue (juniper = -27 ‰; oak = -29 ‰) obtained for these $C_3$ woody species. This isotopic disequilibrium between soils and current organic matter inputs is even more marked in the 10-20 and 20-40 cm depth increments. This suggests that a considerable proportion of SOC beneath juniper and oak stands was derived from $C_4$ grasses, and that this landscape was once a more open $C_4$-dominated grassland. Mass balance calculations based on plant and soil $\delta^{13}C$ values (Eq. 2) indicate that approximately 45-55% of SOC at 0-10 cm and 70-90% of SOC at 20-40 cm is derived from $C_4$ grass in soils atop both Buda and Edwards limestone (Fig. 3b). It should be noted that the absolute



accuracy of these mass balance calculations could be affected by historical changes in the $\delta^{13}C$ values of atmospheric $CO_2$
(Graven et al., 2017), isotopic fractionation during soil organic matter decay (Mainka et al., 2022), and differential
decomposition of $C_3$ vs. $C_4$ plant tissues (Wynn et al., 2020). However, these uncertainties are not large enough to alter the
conclusion that $C_3$ juniper and oak trees are more recent components of this landscape compared to $C_4$ grasses. Our findings
that woody plants are relatively recent components of this landscape are consistent with other studies on the Edwards Plateau
and other portions of the southern Great Plains region based on direct observations of plant community changes through time
(Briggs et al., 2002; Smeins and Fuhlendorf, 1997; Van Auken, 2008), $\delta^{13}C$ values of SOC (Jessup et al., 2003), and sequential
remote sensing imagery (Wang et al., 2018a).

**4.2 Soil C, N, and P concentrations and densities**

Vegetation cover, geology, and soil depth have interacted to significantly alter the horizontal and vertical distribution
of soil C, N, and P concentrations and densities within the soil profile (Tables 2 and S1; Figs. 4 and S3). The most consistently
significant factor influencing soil C, N and P concentrations and densities was the interaction between vegetation type and
geology. This interaction was due to larger differences in soil C, N, and P between vegetation types on Edwards than on the
Buda formation. Soil organic C, TN, and TP concentrations and densities were generally higher under oak and juniper canopies
than under grasslands within all depth increments (Figs. 4 and S3). In addition, the cumulative amounts of SOC, TN, and TP
present throughout the entire 0-40 cm depth were always greater in soils atop the Edwards that those atop the Buda limestone
for all three vegetation types (Fig. 4), with the exception that TP under juniper was lower in soil atop the Edwards formation.
We speculate that the higher concentrations of C, N, and P in soils atop the Edwards limestone may be due to the shallow
depth to bedrock (approximately 40 cm) that constrains root and litter inputs to a limited soil volume. In contrast, depth to
bedrock on the Buda formation can be > 1 m, providing a significantly greater soil volume in which root and litter inputs can
be distributed, thereby reducing C, N, and P concentrations and densities. However, it should be noted that Ashe juniper roots
have been shown to achieve rooting depths of 10 m and live oak roots can reach 20 m by exploiting cracks and fissures in the
bedrock on the Edwards Plateau (Jackson et al., 1999).

Similar studies in other portions of the southern Great Plains and the western USA have also shown that juniper and/or
oak encroachment into grasslands resulted in higher soil nutrient storage (Fernandez et al., 2013; Jessup et al., 2003; McKinley
and Blair, 2008; Shawver et al., 2018). Furthermore, our results are broadly consistent with prior studies around the world
showing that tree/shrub encroachment into grasslands, savannas, deserts, and other arid/semiarid ecosystems generally results
in increased concentrations and pools sizes of soil C, N, P, and other essential elements (Archer et al., 2017; Barger et al.,
2011; Blaser et al., 2014; Eldridge et al., 2011; Zhou et al., 2017, 2018b, 2021).

The increases in soil C, N, and P stores under woody canopies following their invasion into previously grass
dominated areas is likely a consequence of multiple factors. First, rates of aboveground net primary productivity in woody
plant encroached areas can be as much as 400 % higher than those in adjacent open grasslands at sites across the Great Plains
and western North America (Knapp et al., 2008). Although there is currently no comparative data on rates of primary



production in open grasslands vs. juniper-oak encroached portions of the Edwards Plateau region, rates of gross primary productivity have been shown to be 55 % higher following juniper invasion compared to open grasslands across the state of Oklahoma (Wang et al., 2018b). Second, root biomass densities in the top 40 cm of the soil profile in our study area were

approximately 1.7 times larger under juniper and oak patches than under grasslands on both Edwards and Buda limestone (Fig. 2c). And third, above- and belowground woody plant tissues have been shown to be biochemically more recalcitrant than grass tissues due to their higher concentrations of suberin, cutin, and syringyl- and vanillyl-lignin subunits (Filley et al., 2008; Boutton et al., 2009). In addition, juniper tissues contain high concentrations of terpenoids (Adams et al., 2013) and oak tissues are enriched in tannins (Mole, 1993), and these secondary compounds have the potential to limit the activity of decomposer

organisms in the soil environment (Hättenschwiler and Vitousek, 2000). Collectively, the higher rates of above- and belowground organic matter inputs coupled with the biochemical recalcitrance of organic matter derived from woody plants likely favors soil organic matter accumulation and larger stores of soil C, N, and P beneath oak and juniper canopies.

Soil inorganic carbon densities ranged from approximately 10-50 kg C m$^{-3}$ across all samples (Fig. 2d), and comprised on average approximately 40 % of the soil total carbon (i.e., SIC + SOC) stores (Figs. 2d and 4a). These values are consistent

with prior assessments of SIC in this region (Smith et al., 2014). The SIC densities were significantly influenced by the vegetation x geology x depth interaction, with differences between the vegetation types being more accentuated atop Edwards limestone, and with values under grassland declining with depth at a more rapid rate than those under woody plants. The SIC densities were generally lower beneath oak and juniper compared to grassland areas (Fig. 2d). This is consistent with previous studies showing that woody plant encroachment into grasslands leads to significant reductions in SIC via soil acidification

(Liu et al., 2020). We suggest that the lower SIC densities beneath woody plant canopies may be a consequence of larger pools of SOC and higher root densities under oak and juniper, which generate higher concentrations of soil $CO_2$ via organic matter decay and root respiration. Those higher soil $CO_2$ concentrations would favor carbonic acid ($H_2CO_3$) production, which would react with calcium carbonate ($CaCO_3$) to form bicarbonate ($HCO_3^-$) which can then dissociate in soil solution and release $CO_2$ from the soil (Ramnarine et al., 2012; Wilsey et al., 2020; Hong and Chen, 2022). Given that SIC is among the largest pools

in the global carbon cycle (940 Pg C, Eswaran et al., 2000), and that SIC is particularly abundant in arid and semiarid regions where woody plant encroachment is prevalent, SIC loss to the atmosphere following encroachment may be an important impact on the carbon cycle (Liu et al., 2020) and should be considered when evaluating the biogeochemical consequences of land cover changes in these regions.

**4.3 Soil stoichiometry**

Ecosystem C, N, and P cycles are strongly coupled through the processes of primary productivity, respiration, and decomposition such that a change in the abundance of one element generally results in proportional changes in the other two elements. However, because the P cycle also has a significant inorganic geochemical component, the behavior of P in the soil environment can become less tightly coupled to that of C and N, resulting in shifts in stoichiometric ratios which can have important consequences for ecosystem structure and function (Finzi et al., 2011). In this study, soil C:N ratios were affected



by the vegetation x geology interaction (Fig. 5a, Table 2). On the Buda limestone, soils beneath grass and oak had higher C:N ratios (12-13) than those under juniper (10-11) throughout the profile.  On the Edwards formation, soil C:N ratios were higher under juniper and oak (12-13) than under grasslands (11). These differences are likely a response to the C:N ratios of the dominant plant species. Prior studies at this site and elsewhere around the western USA have shown that C:N ratios of leaves and roots of woody plants (oak species = 29-34; juniper species = 43-52) are higher than those of grasses (22-23) (Berner and

Law, 2016; Marshall, 1995; Pregitzer et al., 2002). Thus, increased soil C and N storage beneath woody plants may be a consequence not only of their biochemical composition (as discussed earlier), but also of their more recalcitrant elemental composition.

Soil C:P and N:P ratios were influenced by vegetation cover, geology, and soil depth (Figs. 5b and c, Table 2). These ratios were generally lowest in grassland soils, indicating that woody encroachment increased SOC and TN relatively more

than TP. Similar changes to soil C:N:P stoichiometry have been documented following woody encroachment in a subtropical savanna in southern Texas (Zhou et al., 2018a). Soil C:P and N:P ratios were also higher in soils atop the Edwards formation compared to those on the Buda limestone, likely due to higher root biomass densities in the Edwards soils (Fig. 2c) which elevate soil C and N inputs. It is unclear why soil TP does not increase proportionally along with soil SOC and TN where woody plants have encroached. However, ecosystems are generally P-limited and have relatively low P inputs from

atmospheric deposition (Mahowald et al., 2008) and weathering of parent material (Prietzel et al., 2022), and these constraints may limit the ability of plants to acquire P in the same relative proportion as they acquire C and N.

## 5 Conclusions

Woody plant encroachment into grasslands, savannas, and other semiarid and arid ecosystems is a globally extensive land cover change that generally alters the quantity and quality of organic matter inputs to the soil, modifying pool sizes and

process rates of the major biogeochemical cycles. We examined how *Juniperus ashei* and *Quercus virginiana* encroachment into mixed grass prairie on the Edwards Plateau of central Texas influenced soil C, N, and P concentrations (g kg$^{-1}$ soil), densities (kg m$^{-3}$ soil), and their stoichiometric relationships. We also assessed the potential role of geology by evaluating these response variables in soils lying atop two different limestone parent materials – the Buda vs. Edwards formations.  Stable carbon isotope ratios ($\delta^{13}$C) of soil organic matter revealed that approximately 45-90 % of soil carbon in the 0-40 cm depth

interval beneath juniper and oak stands was derived from C$_4$ plants, confirming that these woody plants were recent components of the landscape. Vegetation cover and geology interacted significantly to change soil C, N, and P. In general, concentrations and densities of all three of these elements were significantly higher under oak and juniper canopies than under grasslands. In addition, C, N, and P concentrations were higher under all three vegetation types in soils derived from the Edwards formation compared to those derived from the Buda formation. We speculate that the higher concentrations of C, N,

and P in soils atop the Edwards limestone may be due to the shallow depth to bedrock (approximately 40 cm) that constrains root and litter inputs to a more limited soil volume.  In contrast, depth to bedrock on the Buda formation is often > 1 m,



providing a significantly greater soil volume in which root and litter inputs could be distributed, thereby resulting in lower C, N, and P concentrations and densities. Soil C:N, C:P, and N:P ratios were generally higher under woody plant canopies compared to grasslands, and higher on soils atop the Edwards formation compared to the Buda formation. The lower ratios in grassland soils indicate that woody encroachment increased SOC and TN relatively more than TP. Our results also show that C and N increase proportionally following woody plant encroachment, likely due to their strong coupling during the processes of primary production, respiration, and decomposition. Although P is also strongly linked to C and N via biological processes, it is also controlled by geochemical processes that can cause its behavior to deviate from that of C and N. Our results are broadly consistent with prior studies around the world showing that tree/shrub encroachment into grasslands, savannas, deserts, and other arid/semiarid ecosystems generally results in increased concentrations and pools sizes of soil C, N, and P, as well as changes in their stoichiometric relationships. Our study also suggests that the magnitude of these changes may be influenced by attributes of the geological formations that underly the soils. Given the geographic extent of woody encroachment at the global scale, our results have important implications for the management and conservation of these ecosystems. We suggest that interactions between vegetation change and geology warrant consideration in future studies, and could play a role in efforts aimed at improving the prediction and modeling of soil C, N, and P storage in grasslands, savannas, and other dryland ecosystems.

**Data availability**

Requests for data and R scripts can be addressed to C.J. Hsiao (chsiao@umn.edu).

**Author contributions**

T. Boutton developed the research concepts. P. Leite surveyed and excavated the soil trenches. C.J. Hsiao conducted the field work. A. Hyodo and C.J. Hsiao carried out soil C, N, and $\delta^{13}$C analyses in the Stable Isotopes for Biosphere Sciences Lab at Texas A&M University. C.J. Hsiao carried the other lab work statistical analysis, data interpretation, and writing with contributions from T. Boutton, P. Leite, and A. Hyodo.

**Competing interests**

The authors declare that they have no conflict of interest.

**Acknowledgments**

Research was supported by the Texas A&M University Sid Kyle Global Savanna Research Initiative (513375-41060), and USDA/NIFA Hatch Project 1020427. We thank Drs. Butch Taylor and Doug Tolleson for providing background information



on land use history and logistical support at the Texas A&M AgriLife Research Station at Sonora, and Dr. Lifei Sun for
assistance with fieldwork.

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
