# Peer review of "Soil carbon, nitrogen, and phosphorus storage in juniper-oak savanna: Role of vegetation and geology"

_EGUsphere, 2023_

## Author Comment (AC1)

**General comments:** The manuscript titled "Soil carbon, nitrogen, and phosphorus storage in juniper-oak savanna: Role of vegetation and geology" is a well-written manuscript that explores how geological factors may interact with woody plant encroachment to influence soil C, N, and P biogeochemistry. The importance of this study is twofold: 1) climate change and certain types of land management are accelerating woody encroachment into grasslands and it's important that we understand how this shift influences soil properties; and 2) it's critical that we understand how geology and vegetation changes interact, as findings can help improve modelling efforts for soil C, N, and P dynamics in various ecosystems going forward. The authors present clear figures and include site photos, which made for a pleasurable read.

**Specific comments:**

**Intro** – Well written! May be helpful to include a brief explanation of how the interaction between soil depth and δ13C of SOC can inform us on the history of the landscape (i.e., Fig. 3a). It would help set up your hypotheses, results, and beginning of your discussion section nicely for those who are less familiar with this concept.

Thank you for your suggestions. I propose to add these sentences into L86:

… grass vs. woody sources of soil organic matter. Grass species in the Edwards Plateau generally use $C_4$ photosynthesis, resulting in different $\delta^{13}C$ values compared to the encroaching woody species, which use $C_3$ photosynthesis (Boutton et al., 1998). Changes in the $\delta^{13}C$ signatures at different soil depths can reveal historical shifts in predominant vegetation, possibly due to climatic changes, disturbances, or human activities (Jessup et al., 2003; Zhou et al., 2019). We tested the hypotheses that…

**Line 109** – can you specify if this higher clay content is across the entire soil profile or across some specific depth?

Thank you for bringing this to our attention. To clarify, I propose modifying the sentence as:

"The clay content in Buda soil is generally 2-5 % lower than that in Edwards soil throughout 0-40 cm profile based on USDA/NRCS data".

**Fig 1** – Great idea to include photos of the Edwards and Buda soils. I think it's helps readers better understand the differences between them (i.e., depth).

Thank you!

**Lines 127 – 129** - Would be useful to add % forage utilization in parentheses here for context on how 'light grazing' vs. 'heavy to moderate grazing' is being defined.

Thank you for your suggestions. I propose to adjust the sentence as:

"Grazing in this area was heavy (2 - 5 ha/animal unit/yr) to moderate (6 - 8 ha/animal unit/yr) from approximately 1880 to 2010, but the area chosen for this study was grazed lightly (9 - 15 ha/animal unit/yr) and intermittently for the past 10 yrs (Leite et al., 2020). Since 1948, the livestock composition in these grazed areas has maintained an approximate ratio of 60:20:20 for cattle, sheep, and goats, respectively (Marshall, 1995)".

**Lines 138 – 139** - Please extrapolate/clarify what you mean by 'within the middle of each depth increment'. I believe you mean 5 cm in the 0-10 cm increment – if you specify this, it will align with the figures better.

Thank you for your suggestions. I suggested to modify the sentence as:

"… We aimed for the midpoint within each specified depth range for sampling. For instance, samples were taken at 5 cm for the 0-10 cm range, 15 cm for the 10-20 cm range, 30 cm for the 20-40 cm range, and so forth. Each sampling point involved the horizontal insertion of two soil cores (7.6 cm width x 10 cm length) into the trench face."

**Table 1 & Table 2** – please consider removing lines between rows in the tables. If this is required formatting, then ignore. Otherwise, I suggest removing and formatting according to journal requirements.

Thank you for your suggestions! I have removed the inside horizontal lines.

**Lines 172 – 173** – The way this is written throws the reader off a little bit. Please consider rewriting as: "The fraction, (*f*), was the proportion of SOC derived . . ." or something similar.

Thank you for your suggestions. I will modify the sentence as:

The fraction, (*f*), was the proportion of SOC derived from $C_4$ plants and $(1 - f)$ is the proportion of SOC derived from $C_3$ plants.

**Lines 201 – 202** – If you weren't able to get BD measurements >20 cm for Edwards soils, how were you able to accurately make SOC predictions past 20 cm (Fig. 4a)? From what I recollect, von Haden requires BD for the input sheet and R script.

This is a great question. Indeed we encountered limitations in getting bulk density for Edwards soil beyond 20 cm depth. Thus, we used a cubic-spline extrapolation method to estimate cumulative SOC stocks from 0-40 cm soil layer based on data from 0-10 and 10-20 cm (Wendt and Hauser, 2013). Then, we can get SOC stock within 20-40 cm soil layer based on the difference between cumulative SOC stocks from 0-20 and 0-40 cm soil layers.

**Lines 208 – 209** – can you clarify what you mean by . . . "the fact that SIC increased more strongly with soil depth beneath oak than beneath grassland or juniper vegetation". Is this based on the slope of the lines in fig 2d? And is it pertaining to across all depths? Just glancing at the figure, it appears the biggest change in SIC between the first and second depth increment is for grass.

Thank you for highlighting this discrepancy. I recognize that there was an error in my initial interpretation presented in the manuscript. I propose revising the sentence on L208 as:

"… the fact that SIC increased more strongly with soil depth beneath grassland than beneath oak or juniper vegetation (Fig. 2d)".

**Table 2 –** Very interesting and surprising that depth alone did not significantly affect SOC. Only the interaction between geology and depth. I suggest capitalizing Depth, Vegetation Geology in the table to make the abbreviations even more intuitive.

Thank you for your suggestions. I will capitalize Depth, Vegetation Geology in Table 2.

**Fig. 3a** – I like the inclusion of δ13C litter values in the same figure as δ13C soil values. However, I would add a statement indicating exactly what the dashed line on the figure indicates in the figure caption for further clarity.

Thank you. I propose the following modification to the captain of Figure 3:

"Changes in (a) $\delta^{13}C$ of litter (above dashed line) and soil, and (b) the fraction of SOC derived from $C_4$ grass beneath juniper and oak in soils derived from the Buda vs. Edwards formations calculated using mass balance. Results are given as means ± standard errors of the mean. Data are plotted at the midpoints of the depth increments."

**Line 244** – perhaps change to … "while **only** oak had higher SOC and TN on Buda soils". It reads a little easier that way.

Thank you. I agree to revise the sentence as:

" Soils beneath live oak and Ashe juniper had higher SOC and TN than grasslands throughout the profile on Edwards soils, while only oak had higher SOC and TN on Buda soils (Figs. 4a and c)."

**Line 306-307** – is it plausible that higher clay content in the Edwards soil could have increased soil C relative to Buda as well? You make a point in the methods that the Buda soil has less clay content.

Thank you. You are correct that differing clay contents between the Edwards and Buda soils could influence the respective carbon storage capacities. I propose revising the sentence on L306-307 as:

We speculate that the higher concentrations of C, N, and P in soils atop the Edwards limestone could be attributed to two factors: First, the higher clay concentration which offers a higher C storage capacity (Basile-Doelsch et al., 2020; Six et al., 2002) and second, the shallow depth to bedrock (approximately 40 cm) in Edwards soil that constrains root and litter inputs to a limited soil volume.

**Discussion** – towards the end of the discussion, it would be helpful to briefly address other ecological effects of woody encroachment that were not directly measured in this study (i.e., biodiversity, soil erosion, etc.). It would make sense to add this sentiment after your point about SIC loss with encroachment (line 347).

Thank you for your suggestion. We recognize the importance of these aspects; however, we believe that including a detailed discussion on topics such as biodiversity and soil erosion might shift the specific focus of our manuscript, which aims to elucidate the interactions between geological factors and woody plant encroachment concerning soil C, N, and P biogeochemistry. Nevertheless, understanding the significance of these points, we propose two potential approaches to acknowledge these important ecological contexts without deviating from our core findings:

1. Within the Discussion section, we could subtly integrate these points as an additional paragraph following section 4.3 Soil stoichiometry. Here's a proposed addition:

   "Recognizing that the altered soil nutrient dynamics have far-reaching ecological consequences, the woody encroachment phenomenon might extend beyond biogeochemical changes. For instance, the modification of microclimatic soil conditions and suppression of herbaceous diversity under woody canopies could influence broader biodiversity (Archer et al., 2017). Furthermore, the disparity in soil nutrient availability between soils under canopies and nutrient-deprived interspaces

(Figs. S1 and S2) may escalate the risk of soil erosion (Puttock et al., 2014; Ravi and D'Odorico, 2009). These considerations, while beyond the primary scope of our current investigation, highlight the multifaceted impacts of woody plant encroachment on arid and semi-arid ecosystems."

2. Alternatively, we could condense these ecological effects and incorporate them into the first paragraph in the Introduction.

Either way, we appreciate your guidance on whether a broader context in the Discussion or a brief mention in the Introduction would be more appropriate, or if another approach might better serve the manuscript.

**Conclusion** – As is, the conclusion is quite long. Please distill and shorten where appropriate – focus on **what** was found and **why** it's important.

Thank you for your suggestions. I suggest to rewrite the Conclusion as:

This study investigated the impact of *Juniperus ashei* and *Quercus virginiana* encroachment on soil C, N, and P stoichiometry in mixed grass prairies on the Edwards Plateau of central Texas, considering the influence of underlying geological variations between soils lying atop two different limestone parent materials – the Buda vs. Edwards formations. Stable C isotope ratios ($\delta^{13}$C) of soil organic matter revealed that 45-90 % of soil C in the 0-40 cm depth interval beneath juniper and oak stands was derived from $C_4$ plants, confirming that these woody plants were recent components of the landscape. Vegetation and geology interaction significantly influenced soil C, N, and P levels, with higher values under juniper and oak canopies than grasslands and on soils derived from the Edwards formation, possibly due to higher clay content and limited soil volume due to shallow depth to bedrock (approximately 40 cm). Conversely, the deeper Buda formation (> 1 m) allowed more extensive root and litter distribution, resulting in lower element concentrations. Soil C:N, C:P, and N:P ratios were generally higher under woody plant canopies compared to grasslands, indicating that woody encroachment increased SOC and TN relatively more than TP. While C and N consistently increased — likely because of their close linkage during primary production, respiration, and decomposition — P trends deviated, reflecting influences from geochemical processes. Our results are broadly consistent with prior studies around the world showing that woody plant encroachment into arid/semiarid ecosystems generally results in increased concentrations and pools sizes of soil C, N, and P, as well as changes in their stoichiometric relationships. Our study also suggests that the magnitude of these changes may be influenced by attributes of the geological formations that underly the soil. Given the geographic extent of woody encroachment at the global scale, our results have important implications for the management and conservation of these ecosystems. We suggest that interactions between vegetation changes and geology warrant consideration in future studies and could play a role in efforts aimed at improving the prediction and modeling of soil C, N, and P storage in grasslands, savannas, and other dryland ecosystems.

---

## Author Comment (AC2)

**Reviewer 1**

**General comments:** The manuscript titled "Soil carbon, nitrogen, and phosphorus storage in juniper-oak savanna: Role of vegetation and geology" is a well-written manuscript that explores how geological factors may interact with woody plant encroachment to influence soil C, N, and P biogeochemistry. The importance of this study is twofold: 1) climate change and certain types of land management are accelerating woody encroachment into grasslands and it's important that we understand how this shift influences soil properties; and 2) it's critical that we understand how geology and vegetation changes interact, as findings can help improve modelling efforts for soil C, N, and P dynamics in various ecosystems going forward. The authors present clear figures and include site photos, which made for a pleasurable read.

**Specific comments:**

**Intro** – Well written! May be helpful to include a brief explanation of how the interaction between soil depth and δ13C of SOC can inform us on the history of the landscape (i.e., Fig. 3a). It would help set up your hypotheses, results, and beginning of your discussion section nicely for those who are less familiar with this concept.

Thank you for your suggestions. I propose to add these sentences into L86:

… grass vs. woody sources of soil organic matter. Grass species in the Edwards Plateau generally use $C_4$ photosynthesis, resulting in different $\delta^{13}C$ values compared to the encroaching woody species, which use $C_3$ photosynthesis (Boutton et al., 1998). Changes in the $\delta^{13}C$ signatures at different soil depths can reveal historical shifts in predominant vegetation, possibly due to climatic changes, disturbances, or human activities (Jessup et al., 2003; Zhou et al., 2019). We tested the hypotheses that…

**Line 109** – can you specify if this higher clay content is across the entire soil profile or across some specific depth?

Thank you for bringing this to our attention. To clarify, I propose modifying the sentence as:

"The clay content in Buda soil is generally 2-5 % lower than that in Edwards soil throughout 0-40 cm soil profile based on USDA/NRCS data".

**Fig 1** – Great idea to include photos of the Edwards and Buda soils. I think it's helps readers better understand the differences between them (i.e., depth).

Thank you!

**Lines 127 – 129** - Would be useful to add % forage utilization in parentheses here for context on how 'light grazing' vs. 'heavy to moderate grazing' is being defined.

Thank you for your suggestions. I propose to adjust the sentence as:

"Grazing in this area was heavy (2 - 5 ha/animal unit/yr) to moderate (6 - 8 ha/animal unit/yr) from approximately 1880 to 2010, but the area chosen for this study was grazed lightly (9 - 15 ha/animal unit/yr) and intermittently for the past 10 yrs (Leite et al., 2020). Since 1948, the livestock composition in these grazed areas has maintained an approximate ratio of 60:20:20 for cattle, sheep, and goats, respectively (Marshall, 1995)".

**Lines 138 – 139** - Please extrapolate/clarify what you mean by 'within the middle of each depth increment'. I believe you mean 5 cm in the 0-10 cm increment – if you specify this, it will align with the figures better.

Thank you for your suggestions. I suggested to modify the sentence as:

"… We aimed for the midpoint within each specified depth range for sampling. For instance, samples were taken at 5 cm for the 0-10 cm range, 15 cm for the 10-20 cm range, 30 cm for the 20-40 cm range, and so forth. Each sampling point involved the horizontal insertion of two soil cores (7.6 cm width x 10 cm length) into the trench face."

**Table 1 & Table 2** – please consider removing lines between rows in the tables. If this is required formatting, then ignore. Otherwise, I suggest removing and formatting according to journal requirements.

Thank you for your suggestions! I have removed the inside horizontal lines.

**Lines 172 – 173** – The way this is written throws the reader off a little bit.  Please consider rewriting as: "The fraction, ($f$), was the proportion of SOC derived . . ." or something similar.

Thank you for your suggestions. I will modify the sentence as:

The fraction, ($f$), was the proportion of SOC derived from $C_4$ plants and $(1 – f)$ was the proportion of SOC derived from $C_3$ plants.

**Lines 201 – 202** – If you weren't able to get BD measurements >20 cm for Edwards soils, how were you able to accurately make SOC predictions past 20 cm (Fig. 4a)? From what I recollect, von Haden requires BD for the input sheet and R script.

This is a great question. Indeed we encountered limitations in getting bulk density for Edwards soil beyond 20 cm depth. Thus, we used a cubic-spline extrapolation method to estimate cumulative SOC stocks from 0-40 cm soil layer based on data from 0-10 and 10-20 cm (L180) (Wendt and Hauser, 2013). Then, we can get SOC stock within 20-40 cm soil layer based on the difference between cumulative SOC stocks from 0-20 and 0-40 cm soil layers.

**Lines 208 – 209** – can you clarify what you mean by . . . "the fact that SIC increased more strongly with soil depth beneath oak than beneath grassland or juniper vegetation". Is this based on the slope of the lines in fig 2d? And is it pertaining to across all depths? Just glancing at the figure, it appears the biggest change in SIC between the first and second depth increment is for grass.

Thank you for highlighting this discrepancy. I recognize that there was an error in my initial interpretation presented in the manuscript. I propose revising the sentence on L208 as:

"… the fact that SIC at the 15 cm and 30 cm depths was substantially higher than at the 5 cm depts beneath grassland than beneath oak or juniper vegetation (Fig. 2d)".

**Table 2 –** Very interesting and surprising that depth alone did not significantly affect SOC. Only the interaction between geology and depth. I suggest capitalizing Depth, Vegetation Geology in the table to make the abbreviations even more intuitive.

Thank you for your suggestions. I will capitalize Depth, Vegetation Geology in Table 2.

**Fig. 3a** – I like the inclusion of δ13C litter values in the same figure as δ13C soil values. However, I would add a statement indicating exactly what the dashed line on the figure indicates in the figure caption for further clarity.

Thank you. I propose the following modification to the captain of Figure 3:

"Changes in (a) $\delta^{13}C$ of litter (above dashed line) and soil, and (b) the fraction of SOC derived from $C_4$ grass beneath juniper and oak in soils derived from the Buda vs. Edwards formations calculated using mass balance. Results are given as means ± standard errors of the mean. Data are plotted at the midpoints of the depth increments."

**Line 244** – perhaps change to … "while **only** oak had higher SOC and TN on Buda soils". It reads a little easier that way.

Thank you. I agree to revise the sentence as:

" Soils beneath live oak and Ashe juniper had higher SOC and TN than grasslands throughout the profile on Edwards soils, while only oak had higher SOC and TN on Buda soils (Figs. 4a and c)."

**Line 306-307** – is it plausible that higher clay content in the Edwards soil could have increased soil C relative to Buda as well? You make a point in the methods that the Buda soil has less clay content.

Thank you. You are correct that differing clay contents between the Edwards and Buda soils could influence the respective carbon storage capacities. I propose revising the sentence on L306-307 as:

We speculate that the higher concentrations of C, N, and P in soils atop the Edwards limestone could be attributed to two factors: First, the higher clay concentration which offers a higher C storage capacity

(Six et al., 2002; Basile-Doelsch et al., 2020) and second, the shallow depth to bedrock (approximately 40 cm) in Edwards soil that constrains root and litter inputs to a limited soil volume.

**Discussion** – towards the end of the discussion, it would be helpful to briefly address other ecological effects of woody encroachment that were not directly measured in this study (i.e., biodiversity, soil erosion, etc.). It would make sense to add this sentiment after your point about SIC loss with encroachment (line 347).

Thank you for your suggestion. We recognize the importance of these aspects; however, we believe that including a detailed discussion on topics such as biodiversity and soil erosion might shift the specific focus of our manuscript, which aims to elucidate the interactions between geological factors and woody plant encroachment concerning soil C, N, and P biogeochemistry. Nevertheless, understanding the significance of these points, we propose two potential approaches to acknowledge these important ecological contexts without deviating from our core findings:

1. Within the Discussion section, we could subtly integrate these points as an additional paragraph following section 4.3 Soil stoichiometry. Here's a proposed addition:

   "Recognizing that the altered soil nutrient dynamics have far-reaching ecological consequences, the impact of woody encroachment might extend beyond biogeochemical changes. For instance, the modification of microclimatic soil conditions and suppression of herbaceous diversity under woody canopies could influence broader biodiversity (Archer et al., 2017). Furthermore, the disparity in soil nutrient availability between soils under canopies and nutrient-deprived interspaces (Figs. S1 and S2) may escalate the risk of soil erosion in drylands (Puttock et al., 2014; Ravi and D'Odorico, 2009; Wilcox et al., 2022). These considerations, while beyond the primary scope of our current investigation, highlight the multifaceted impacts of woody plant encroachment on arid and semi-arid ecosystems."

2. Alternatively, we could condense these ecological effects and incorporate them into the first paragraph in the Introduction.

Either way, we appreciate your guidance on whether a broader context in the Discussion or a brief mention in the Introduction would be more appropriate, or if another approach might better serve the manuscript.

**Conclusion** – As is, the conclusion is quite long. Please distill and shorten where appropriate – focus on **what** was found and **why** it's important.

Thank you for your suggestions. I suggest to rewrite the Conclusion as:

This study investigated the impact of *Juniperus ashei* and *Quercus virginiana* encroachment on soil C, N, and P stoichiometry in mixed grass prairies on the Edwards Plateau of central Texas, considering the influence of underlying geological variations between soils lying atop two different limestone parent materials – the Buda vs. Edwards formations. Stable C isotope ratios ($\delta^{13}$C) of soil organic matter revealed that 45-90 % of soil C in the 0-40 cm depth interval beneath juniper and oak stands was derived

from C4 plants, confirming that these woody plants were recent components of the landscape. Vegetation and geology interaction significantly influenced soil C, N, and P levels, with higher values under juniper and oak canopies than grasslands and on soils derived from the Edwards formation, possibly due to higher clay content and limited soil volume due to shallow depth to bedrock (approximately 40 cm). Conversely, the deeper Buda formation (> 1 m) allowed more extensive root and litter distribution, resulting in lower element concentrations. Soil C:N, C:P, and N:P ratios were generally higher under woody plant canopies compared to grasslands, indicating that woody encroachment increased SOC and TN relatively more than TP. While C and N consistently increased — likely because of their close linkage during primary production, respiration, and decomposition — P trends deviated, reflecting influences from geochemical processes. Our results are broadly consistent with prior studies around the world showing that woody plant encroachment into arid/semiarid ecosystems generally results in increased concentrations and pools sizes of soil C, N, and P, as well as changes in their stoichiometric relationships. Our study also suggests that the magnitude of these changes may be influenced by attributes of the geological formations that underly the soil. Given the geographic extent of woody encroachment at the global scale, our results have important implications for the management and conservation of these ecosystems. We suggest that interactions between vegetation changes and geology warrant consideration in future studies and could play a role in efforts aimed at improving the prediction and modeling of soil C, N, and P storage in grasslands, savannas, and other dryland ecosystems.

**Reviewer 2**

**General Comments**

This study quantified soil properties by depth (SOC, δ13 C, nitrogen, and phosphorus) under contemporary ecological conditions (grassland, juniper, and oak) on different ecological sites (by depth and parent material; Edwards, Buda) to evaluate impact on the soil's biogeochemistry. Results show that grass to woodland transition is relatively recent, and that vegetation transition dynamics and soil parent material uniquely condition soil nutrient stores. My main concern regards conclusions related to land use change (grass to shrub) absent of long-term data (quantification of ecological state change over time) or a more specific soil-chronosequence study (space for time substitution, better control of soil type and soil age). Here are a few suggestions for the authors to consider on revision.

The presence of a petrocalcic horizon in the Prade (Edwards) and Valera (Buda) soils infer these soils are pedogenically much older than the Eckrant (Edwards) and Tarrant (Buda) soils. Valera also has a different soil family texture class (fine) than the other three (clayey skeletal). Can the authors provide the soil taxonomy and geographic locations for the soil trenches in Table 1? This will significantly add the soil and landscape interpretation. Without this information, "shallow depth to bedrock" (L385) for the Prade and Valera soils could be confused with the "depth to petrocalcic horizon". Petrocalcic horizons are considered 'pedogenic' (atmospheric additions with soil translocations and transformations) and limestone/marl is older, or 'geogenic'. Without these data, authors could maintain some their assumptions of soil behavior (shallow vs deep), however, conclusions of geology's role are more complicated.

Thank you for your insightful comments. I propose the following modifications to the soil pedology section 2.1 and Table 1. Please note that although the profiles of some sampling locations within Trench 1, 2, 5A, 5B, and 6 have been described and published in Soil Pedon Description (attached as appendix to this Reply to Reviewer document), the extensive length of some trenches indicated the presence of multiple soil series. Thus, we presented soil map units instead of soil series in Table 1 for clarity. In our revised paragraph, we provide both the depth to bedrock and depth to caliche (petrocalcic or paralithic) horizon. It is worth noting that the caliche layers in Trench 5A, 5B, and 6 were unconsolidated, and root growth was observed within the caliche layers.

Proposed revision of pedology paragraph (L98):

The Edwards Plateau is an uplifted and dissected limestone plateau (karst topography) with gentle slopes. Soils in this region are clayey Mollisols with shallow soils on plateaus and hills, and deeper soils on plains and valley floors (Gabriel et al., 2009; Wiedenfeld and McAndrew, 1968). The predominant soil map units on plateaus and hills are the Eckrant – Rock outcrop complex and the Prade – Eckrant complex. These include the commonly occurred Harper (clayey, smectitic, thermic Lithic Haplustolls), Prade (Clayey-skeletal, smectitic, thermic, shallow Petrocalcic Calciustolls), and Tarrant soil series (clayey-skeletal, smectitic, thermic Lithic Calciustolls) , all of which lie atop the Edwards formation (Wilcox et al., 2007). These soils contain large amounts of limestone fragments and limestone outcrops. Depth to bedrock for these soils was generally < 0.4 m (Figs. 1 and S1). The clay content in the top 5 cm of Edwards soil is 30-40% and increases with depth to almost 50% at 20 cm depth (Marshall, 1995). The soil map unit commonly occurring on plains and valley floors is Valera clay, including Rio Diablo (Fine, mixed, superactive, thermic Aridic Haplustolls), Ozona (Loamy, mixed, superactive, thermic, shallow Petrocalcic Calciustolls), and Mereta (Clayey, mixed, superactive, thermic, shallow Petrocalcic Calciustolls), which lie atop the Buda formation (Wilcox et al., 2007; Gabriel et al., 2009). These soils are generally deeper than those lying atop the Edwards formation and contain hard limestone and a caliche layer on top of the limestone bedrock (Figs. 1 and S1). The caliche layer, typically light in color, can manifest as either the petrocalcic (Bkkm horizon) or the paralithic (Cr) layer, both potentially cemented or unconsolidated. The petrocalcic layer is comprised of weathered carbonate and cements the B horizon with soil particles (Soil Survey Staff, 2014), while the paralithic layer is a residuum from limestone weathering. The clay content in Buda soil is generally 2-5 % lower than that in Edwards soil throughout 0-40 cm profile based on USDA/NRCS data (Official Soil Series Descriptions (Rio Diablo series), 2022; Official Soil Series Descriptions (Eckrant series), 2022; Official Soil Series Descriptions (Valera series), 2022; Official Soil Series Descriptions (Prade series), 2022). Depth to consolidated bedrock for Buda soils was approximately 1.5-2 m, while depth to the caliche layer was approximately 0.5 m. Although the hard limestone geological formations underlying both the Edwards and Buda soils have contributed somewhat to the formation of these soils, there is considerable chemical and physical evidence indicating that these soils are derived largely from an overlying limestone residuum with distinctly different attributes than the underlying limestone (Rabenhorst and Wilding, 1986a, b; Cooke et al., 2007). The Del Rio Clay, an Upper Cretaceous marly limestone that locally overlies the Edwards limestone, has been proposed as the dominant source of these soils (based on texture, mineralogy, and Nd isotope composition), at least on the eastern portion of the Edwards Plateau (Cooke et al., 2007).

Table 1. Geological substrates and physical dimensions of soil trenches.

| Trench | Geological formation | Average depth (m) | Length (m) | Soil series | Landscape | Coordinates |
|---|---|---|---|---|---|---|
| 1 | Edwards | 0.7 | 30 | Haper – Prade | Footslope | 30°17'22.6"N, 100°33'33.0"W |
| 2 | Edwards | 0.4 | 8 | Tarrant | Summit | 30°17'22.4"N, 100°33'12.1"W |
| 4 | Edwards | 0.4 | 14 | Tarrent | Summit | 30°16'48.4"N, 100°33'37.8"W |
| 5A | Buda | 1.5 | 20 | Prade – Ozona | Footslope | 30°15'20.1"N, 100°34'21.5"W |
| 5B | Buda | 1.4 | 9 | Rio Diablo | Toeslope | 30°15'18.9"N, 100°34'19.8"W |
| 6 | Buda | 1.4 | 12 | Mereta – Rio Diablo | Toeslope | 30°17'00.0"N, 100°32'27.9"W |

**Specific Comments:**

L21, L398, "results have important implications for the management and conservation of these ecosystem". Can the authors add to the discussion management suggestions and implications?

We proposed a modification in L402:

Our study also suggests that the magnitude of these changes may be influenced by attributes of the geological formations that underly the soils. Given the geographic extent of woody encroachment at the global scale, our results suggest that soil conservation practices might need to be tailored according to the underlying geology. Interactions between vegetation change and geology warrant consideration in future studies, and could play a role in efforts aimed at improving the prediction and modeling of soil C, N, and P storage in grasslands, savannas, and other dryland ecosystems.

L85, is the first mention of δ13 C to test woody encroachment. Given that δ13 C is used to conclude grassland to woodland conversion (L381), consider adding to the introduction how carbon isotopes are used as proxy to identify relative abundance C3 and C4 vegetation.

Thank you for your suggestions. I propose to add these sentences into L86:

… grass vs. woody sources of soil organic matter. Grass species in the Edwards Plateau generally use $C_4$ photosynthesis, resulting in different $δ^{13}C$ values compared to the encroaching woody species, which use $C_3$ photosynthesis (Boutton et al., 1998). Changes in the $δ^{13}C$ signatures at different soil depths can reveal historical shifts in predominant vegetation, possibly due to climatic changes, disturbances, or human activities (Jessup et al., 2003; Zhou et al., 2019). We tested the hypotheses that…

L101, L104, include the taxonomy for each soil class. Eckrant is "Clayey-skeletal, smectitic, thermic Lithic Haplustolls"; Prade is "Clayey-skeletal, smectitic, thermic, shallow Petrocalcic Calciustolls"; Valera is "Fine, smectitic, thermic Petrocalcic Calciustolls"; Tarrant is "Clayey-skeletal, smectitic, thermic Lithic Calciustolls".

Thank you for your suggestions. Soil taxonomy were provided in the proposed revision. Please refer to the response to general comments.

L104, Valerna should be Valera.

Thank you for pointing out this mistake. I have corrected it.

L110, is the only mention of the Rio Diablo and Ector soil series. Are these series identified in the study? Do these soil series add any additional information to the study (Buda vs Edward)?

Thank you for your suggestions. Rio Diablo is one of the major soils atop Buda limestone. For example, trench 6 is located in Rio Diablo silty clay soil mapping unit. Ector soil series is a commonly observed soil lying atop Edwards limestone in Edwards Plateau (Gabriel et al., 2009). As none of the trenches in this study were located in an Ector-related map unit, we agree to remove Ector soils from the references for clarity.

L107, Bkkm is a "petrocalcic horizon" (pedogenic), avoid calling it marl (geogenic).

Thank you for your correction. Please note that the caliche layers in soil profiles of Trench 5 and 6 (both Buda limestone formation) is different. According to soil pedon descriptions, the yellow-whitish layer of Trench 5 is unconsolidated paralithic (Cr) layer weathered from limestone. The gray fractured layer between A and Cr horizons in Trench 6 is petrocalcic (Bkkm) horizon. Please see the figures below. This interpretation is supported by the findings of a previous study near the Sonora station (Rabenhorst and Wilding, 1986a).

We also propose adding Figure X below to the supplemental materials.

[Figure]

[Figure]

Figure X. Trench 5A (a) and Trench 6 (b) in Buda soil at Texas A&M AgriLife Sonora Research Station on the Edwards Plateau, Texas.

L129, can authors provide more information regarding the historic land cover conversion dynamics at these locations? Such as the historic rate and magnitude of the grassland to woodland conversion in the study area? I suspect that the TAMU Sonora station has this data.

Although we do not have records of grassland to woodland conversion rates at our locations, woody cover was shown to increase from 7.6% to 19.3% between 1986 and 2020 in other portions of the Sonora station, corresponding to average annual increase of 0.8% per year (Leite et al., 2023). This rate is comparable to regional observed in South Texas and Oklahoma, where annual increases in juniper cover averaged 0.5-1.5% per year (Barger et al., 2011; Archer et al., 2001; Fowler and Simmons, 2009; Wang et al., 2018).

L136, can you add a simple rational for the depth intervals used in the study.

We proposed to add those sentences into L136:

…particularly those atop the Edwards formation. Soil depth intervals were selected to represent soil horizon patterns described in USDA/NRCS Web Soil Survey and comparability with previous studies of the study area. Within the middle of each depth increment,…

L146, Table 1, can you provide specific site characteristics (landscape position), taxonomic classes, and the geographic coordinates for the trenches? Site characteristics and Soil taxonomy will add to the interpretation, and coordinates will help confirm any soil-landscape relationships previously identified by soil survey.

This is a great suggestion. Please see the proposed revised version of Table 1 as below. Please note that Trench 5 spans the landscape from footslope to toeslope according to the pedon descriptions.

Table 2. Geological substrates and physical dimensions of soil trenches.

| Trench | Geological formation | Average depth (m) | Length (m) | Soil series | Landscape | Coordinates |
|---|---|---|---|---|---|---|
| 1 | Edwards | 0.7 | 30 | Haper – Prade | Footslope | 30°17'22.6"N, 100°33'33.0"W |
| 2 | Edwards | 0.4 | 8 | Tarrant | Summit | 30°17'22.4"N, 100°33'12.1"W |
| 4 | Edwards | 0.4 | 14 | Tarrent | Summit | 30°16'48.4"N, 100°33'37.8"W |
| 5A | Buda | 1.5 | 20 | Prade – Ozona | Footslope | 30°15'20.1"N, 100°34'21.5"W |
| 5B | Buda | 1.4 | 9 | Rio Diablo | Toeslope | 30°15'18.9"N, 100°34'19.8"W |
| 6 | Buda | 1.4 | 12 | Mereta – Rio Diablo | Toeslope | 30°17'00.0"N, 100°32'27.9"W |

L171, I am not familiar with what appears to be a simplified formula to determine the C4 fraction. A more precise formula uses an end member mixing model of two sources: $\%C_3 = [(\delta^{13}C_s - \delta^{13}C_{C4})/(\delta^{13}C_{C3} - \delta^{13}C_{C4})] \cdot 100\%$, and then calculates from, $\%C4 = 100 - \%C3$, (Phillips & Greg, 2001, Oecologia).

These two equations are equivalent:

$\%C_3 = [(\delta^{13}C_s - \delta^{13}C_{C4})/(\delta^{13}C_{C3} - \delta^{13}C_{C4})]$

$(1 − \%C_4) \times (\delta^{13}C_{C3} − \delta^{13}C_{C4}) = \delta^{13}C_s − \delta^{13}C_{C4}$

$\delta^{13}C_s = (1 − \%C_4) \times (\delta^{13}C_{C3} − \delta^{13}C_{C4}) + \delta^{13}C_{C4}$

$\delta^{13}C_s = (1 − \%C_4) \times \delta^{13}C_{C3} − \%C_4 \times \delta^{13}C_{C4}$

Please note that the formula adopted in our study has been developed by previous studies (Jessup et al., 2003; Boutton et al., 1998).

L172, was C4 and C3 tissue collection completed at each trench (such as the litter in Fig 3a) for the mixing model?

For the mixing model, we collected live leaves from three individuals of Ashe juniper, live oak ($C_3$ species), and dominant $C_4$ grass species near each trench. The 100% $C_4$ ($\delta^{13}C_{C4}$) endpoint for grasslands was -16.3 ± 0.8‰, while $\delta^{13}C$ for Ashe juniper and live oak were -26.8 ± 0.4‰ and -28.5 ± 0.5‰, respectively. These values were integral to the mixing model, helping to accurately characterize the contributions of $C_3$ and $C_4$ plants to the soil organic matter pool.

L206, I advise caution interpreting how geology and vegetation impact the SIC properties. Untangling SIC complexity likely requires a detailed soil-chronosequence study with specific controls on soil type (topographic position, genetic horizon, carbonates, etc) and the timing of vegetation change.

Thank you for emphasizing the need for cautious interpretation when examining the relationship between geology, vegetation, and SIC properties. We agree that specific controls on soil characteristics and the timing of vegetation change would provide more definitive insights into role of geology on SIC complexities. That's why we have focused on the potential impacts of vegetation and depth interaction on SIC in our Discussion (L338-344), avoiding an overinterpretation on the role of geology. In addition, we propose to adjust our manuscript to further align with your recommendation (L344):

… release $CO_2$ from the soil (Ramnarine et al., 2012; Wilsey et al., 2020; Hong and Chen, 2022). We acknowledge that factors such as topographic position and the presence of carbonate-rich horizons may also contribute to complex interactions between soil depth, vegetation, and geology in shaping SIC profiles on Edwards and Buda soils. Our interpretations of SIC profile were based on the assumption that the extent and timing of woody encroachment are consistent across soils stop Edwards and Buda limestones in Edwards Plateau. Given that SIC is among the largest pools in the global carbon cycle…

L208, Fig 2d, I wonder if SIC increase is due to the soils being inherently different (Petrocalcic Calciustolls vs Lithic Haplustolls). The Grassland-Edwards site has a considerable amount more SIC than the other sites.

Thank you for this possible explanation. However, both Calciustolls and Haplustolls are commonly observed in soils atop Edwards and Buda limestone formation. For example, Trench 1 located in the Eckrant-Rock outcrop complex, spanning across the Harper (Haplustolls) and Prade (Calciustolls) soil series. Similarly, Trench 5 includes a mix of soil series: Campwood and Rio Diablo (Haplustolls) as well as

Prade and Ozona (Calciustolls). This diverse soil composition across our study sites suggests that the variations in SIC are not solely attributable to the binary classification of Calciustolls vs. Haplustolls. Other factors, potentially including microclimate variations, vegetation, or organic matter contributions may also play significant roles in shaping SIC levels in these regions.

Figure 3b, any reason why Edwards-Grass is not part of figure 3b?

Thank you for pointing this out. Because the objective of Fig. 3b is to show the fraction of SOC derived from C4 grass beneath juniper and oak, neither of the Buda nor Edwards graph should include grass vegetation. Here is the correct figure:

[Figure]

L342-344, Yes, but does this generate enough carbonic acid to alter (within the timeline of land type conversion) the petrocalcic horizon? I advise caution interpreting SIC results without knowing the presence/ absence of pedogenic carbon (calcic, petrocalcic) vs geogenic carbon (limestone/ marl).

Both Buda and Edwards soils in Edwards Plateau are rich in carbonates. As you pointed out, the difference in carbonate weathering rates between woodlands and grasslands may not significant under the assumption of consistent water flow partitioning across vegetation types (Wen et al., 2021). However, at soils atop Buda limestone, it has been observed woody encroachment contributed to a 24–44% increase in weathered limestone (Cr horizon) porosity within less than a century (Leite et al., 2023). This change could be attributed to deeper roots facilitating greater infiltration rates, which might enhance bicarbonate exportation and alter the dissolution equilibrium (Leite et al., 2023; Wen et al., 2021).

We proposed to add this into L 344:

[revised manuscript text omitted]

---

## Author Comment (AC4)

**PEDON DESCRIPTION (Trench 1, location 5 m)**

**Print Date:** Jun 11 2019
**Description Date:** Mar 13 2019
**Describer:** Ashley Anderson, Travis Waiser, Geraldine Vega
**Site ID:** S2019TX1370007

**Pedon ID:** S2019TX1370007

**Site Note:**

**Pit Location:**
**Pedon Note:**

**Lab Source ID:**
**Lab Pedon #:**

**User Transect ID:**
**Soil Name as Described/Sampled:** Harper
**Classification:** Clayey, smectitic, thermic Lithic Haplustolls

**Soil Name as Correlated:**

**Classification:**
**Pedon Type:** undefined observation
**Pedon Purpose:** research site

**Taxon Kind:** family

**Associated Soils:**
**Physiographic Division:**
**Physiographic Province:**
**Physiographic Section:**

**State Physiographic Area:**

**Local Physiographic Area:**
**Geomorphic Setting:** on footslope of base slope of ridge on dissected plateau
**Upslope Shape:** concave
**Cross Slope Shape:** linear

**Country:**
**State:** Texas
**County:** Edwards
**MLRA:** 81B -- Edwards Plateau, Central Part
**Soil Survey Area:** TX607 -- Edwards and Real Counties, Texas
**Soil Survey Area:** TX607 -- Edwards and Real Counties, Texas

**Map Unit:**
**Quad Name:** Dunbar Draw SE, Texas
**Std Latitude:** 30.2895833
**Std Longitude:** -100.5594333

**Latitude:** 30 degrees 17 minutes 22.50 seconds north
**Longitude:** 100 degrees 33 minutes 33.96 seconds west

**Datum:** WGS84
**UTM Zone:** 14
**UTM Easting:** 350026 meters

**UTM Northing:** 3351904 meters

**Primary Earth Cover:**
**Secondary Earth Cover:**
**Existing Vegetation:** algerita, live oak, redberry juniper, slim tridens, Texas persimmon, Texas wintergrass

**Parent Material:** alluvium derived from limestone

**Bedrock Kind:**

**Bedrock Depth:**

**Bedrock Hardness:**

**Bedrock Fracture Interval:**

**Particle Size Control Section:** 25 to 44 cm.

**Description origin:** NASIS

**Diagnostic Features:** mollic epipedon 0 to 18 cm.
cambic horizon 18 to 44 cm.
lithic contact 44 to cm.

**Surface Fragments:**

**Description database:**
MLRA09_Temple

| Top Depth (cm) | Bottom Depth (cm) | Restriction Kind | Restriction Hardness |
|---|---|---|---|
| 44 | | bedrock, lithic | Indurated |

**Cont. Site ID:** S2019TX1370007

**Pedon ID:**
S2019TX1370007

| Slope (%) | Elevation (meters) | Aspect (deg) | MAAT (C) | MSAT (C) | MWAT (C) | MAP (mm) | Frost-Free Days | Drainage Class | Slope Length (meters) | Upslope Length (meters) |
|---|---|---|---|---|---|---|---|---|---|---|
| | 675.7 | | | | | | | | | |

A--0 to 18 centimeters (0.0 to 7.1 inches); silty clay, very dark brown (10YR 2/2), moist; moderate fine granular structure; slightly hard, friable; common very fine roots throughout and common fine roots throughout; 4 percent nonflat subangular indurated 2 to 75-millimeter Limestone fragments; strong effervescence, by HCl, 1 normal; gradual smooth boundary.

Bw--18 to 44 centimeters (7.1 to 17.3 inches); silty clay, brown (10YR 4/3), moist; weak medium subangular blocky, and moderate fine subangular blocky structure; slightly hard, friable; common very fine roots throughout and common fine roots throughout; 7 percent nonflat subangular indurated 2 to 75-millimeter Limestone fragments; violent effervescence, by HCl, 1 normal; abrupt smooth boundary.

R--44 centimeters (17.3 inches); bedrock; .

**PEDON DESCRIPTION (Trench 1, location 17 m)**

**Print Date:** Jun 11 2019
**Description Date:** Mar 13 2019
**Describer:** Ashley Anderson, Travis Waiser, Geraldine Vega
**Site ID:** P2019TX1370005

**Pedon ID:** P2019TX1370005

**Site Note:**

**Pit Location:**
**Pedon Note:**

**Lab Source ID:**
**Lab Pedon #:**

**User Transect ID:**
**Soil Name as Described/Sampled:** Prade
**Classification:** Clayey-skeletal, smectitic, thermic, shallow Petrocalcic Calciustolls

**Soil Name as Correlated:**

**Classification:**
**Pedon Type:** undefined observation
**Pedon Purpose:** research site

**Taxon Kind:** family

**Associated Soils:**
**Physiographic Division:**
**Physiographic Province:**
**Physiographic Section:**

**State Physiographic Area:**

**Local Physiographic Area:**
**Geomorphic Setting:** on backslope of side slope of ridge on dissected plateau
**Upslope Shape:** linear
**Cross Slope Shape:** linear

**Country:**
**State:** Texas
**County:** Edwards
**MLRA:** 81B -- Edwards Plateau, Central Part

**Soil Survey Area:** TX607 -- Edwards and Real Counties, Texas

**Soil Survey Area:** TX607 -- Edwards and Real Counties, Texas

**Map Unit:**
**Quad Name:** Dunbar Draw SE, Texas

**Std Latitude:** 30.2895667
**Std Longitude:** -100.5593000

**Latitude:** 30 degrees 17 minutes 22.44 seconds north

**Longitude:** 100 degrees 33 minutes 33.48 seconds west

**Datum:** WGS84
**UTM Zone:** 14
**UTM Easting:** 350038 meters

**UTM Northing:** 3351902 meters

**Primary Earth Cover:**
**Secondary Earth Cover:**
**Existing Vegetation:** cedar sedge, hairy wedelia, purple threeawn, redberry juniper, Texas bluebonnet

**Parent Material:** residuum weathered from limestone
**Bedrock Kind:**

**Bedrock Depth:**

**Bedrock Hardness:**

**Bedrock Fracture Interval:**

**Particle Size Control Section:** 0 to 34 cm.

**Surface Fragments:** 25.0 percent nonflat subangular indurated 2- to 75- millimeter Limestone fragments and 15.0 percent nonflat subangular indurated 75- to 250- millimeter Limestone fragments

**Description origin:** NASIS

**Description database:** MLRA09_Temple

**Diagnostic Features:** mollic epipedon 0 to 34 cm.
petrocalcic horizon 34 to 65 cm.
lithic contact 65 to cm.

| Top Depth (cm) | Bottom Depth (cm) | Restriction Kind | Restriction Hardness |
|---|---|---|---|
| 34 | 65 | petrocalcic | Strongly cemented |
| 65 | | bedrock, lithic | Indurated |

**Cont. Site ID:** P2019TX1370005

**Pedon ID:**
P2019TX1370005

| Slope (%) | Elevation (meters) | Aspect (deg) | MAAT (C) | MSAT (C) | MWAT (C) | MAP (mm) | Frost-Free Days | Drainage Class | Slope Length (meters) | Upslope Length (meters) |
|---|---|---|---|---|---|---|---|---|---|---|
| | 676.7 | | | | | | | | | |

A--0 to 14 centimeters (0.0 to 5.5 inches); gravelly clay, black (10YR 2/1), moist; moderate fine granular structure; slightly hard, friable; common very fine roots throughout and few medium roots throughout and common fine roots throughout; 5 percent nonflat subangular indurated 75 to 250-millimeter Limestone fragments and 12 percent nonflat subangular indurated 2 to 75-millimeter Limestone fragments; violent effervescence, by HCl, 1 normal; clear smooth boundary.

Bw--14 to 34 centimeters (5.5 to 13.4 inches); very dark grayish brown (10YR 3/2) very gravelly clay, very dark brown (10YR 2/2), moist; moderate fine granular structure; slightly hard, friable; common very fine roots throughout and common medium roots throughout and common fine roots throughout; 20 percent nonflat subangular indurated 75 to 250-millimeter Limestone fragments and 35 percent nonflat subangular indurated 2 to 75-millimeter Limestone fragments; violent effervescence, by HCl, 1 normal; abrupt wavy boundary.

Bkkm--34 to 65 centimeters (13.4 to 25.6 inches); material; few very fine roots throughout and few fine roots throughout; abrupt wavy boundary.

R--65 centimeters (25.6 inches); bedrock; .

**PEDON DESCRIPTION  (Trench 1, location 25 m)**

**Print Date:** Jun 11 2019
**Description Date:** Mar 13 2019
**Describer:** Ashley Anderson, Travis Waiser, Geraldine Vega
**Site ID:** S2019TX1370004

**Pedon ID:** S2019TX1370004

**Site Note:**

**Pit Location:**
**Pedon Note:**

**Lab Source ID:**
**Lab Pedon #:**

**User Transect ID:**
**Soil Name as Described/Sampled:** Prade
**Classification:** Clayey-skeletal, smectitic, thermic, shallow Petrocalcic Calciustolls

**Soil Name as Correlated:**

**Classification:**
**Pedon Type:** correlates to named soil
**Pedon Purpose:** research site

**Taxon Kind:** series

**Associated Soils:**
**Physiographic Division:**
**Physiographic Province:**
**Physiographic Section:**

**State Physiographic Area:**

**Local Physiographic Area:**
**Geomorphic Setting:** on backslope of side slope of ridge on dissected plateau
**Upslope Shape:** linear
**Cross Slope Shape:** linear

**Country:**
**State:** Texas
**County:** Edwards
**MLRA:** 81B -- Edwards Plateau, Central Part

**Soil Survey Area:** TX607 -- Edwards and Real Counties, Texas
**Soil Survey Area:** TX607 -- Edwards and Real Counties, Texas

**Map Unit:**
**Quad Name:** Dunbar Draw SE, Texas

**Std Latitude:** 30.2895167
**Std Longitude:** -100.5591667

**Latitude:** 30 degrees 17 minutes 22.26 seconds north
**Longitude:** 100 degrees 33 minutes 33.00 seconds west
**Datum:** WGS84
**UTM Zone:** 14
**UTM Easting:** 350051 meters
**UTM Northing:** 3351896 meters

**Primary Earth Cover:**
**Secondary Earth Cover:**
**Existing Vegetation:** cedar sedge, hairy wedelia, redberry juniper, Texas bluebonnet, Texas pricklypear
**Parent Material:** residuum weathered from limestone
**Bedrock Kind:**
**Bedrock Depth:**
**Bedrock Hardness:**
**Bedrock Fracture Interval:**

**Particle Size Control Section:** 0 to 31 cm.

**Surface Fragments:** 30.0 percent nonflat subangular indurated 2- to 75-millimeter Limestone fragments and 34.0 percent nonflat subangular indurated 75- to 250-millimeter Limestone fragments and 1.0 percent nonflat subangular indurated 250- to 600-millimeter Limestone fragments

**Description origin:** NASIS

**Description database:** MLRA09_Temple

**Diagnostic Features:** mollic epipedon 0 to 31 cm.
petrocalcic horizon 31 to 50 cm.
paralithic contact 50 to cm.

| Top Depth (cm) | Bottom Depth (cm) | Restriction Kind | Restriction Hardness |
|---|---|---|---|
| 31 | 50 | petrocalcic | Weakly cemented |
| 50 | | bedrock, paralithic | Weakly cemented |

**Cont. Site ID:** S2019TX1370004

**Pedon ID:** S2019TX1370004

| Slope (%) | Elevation (meters) | Aspect (deg) | MAAT (C) | MSAT (C) | MWAT (C) | MAP (mm) | Frost-Free Days | Drainage Class | Slope Length (meters) | Upslope Length (meters) |
|---|---|---|---|---|---|---|---|---|---|---|
| | 679.1 | | | | | | | | | |

A--0 to 15 centimeters (0.0 to 5.9 inches); clay, black (10YR 2/1), moist; moderate fine granular structure; slightly hard, friable; common very fine roots throughout and common medium roots throughout and common fine roots throughout; 2 percent nonflat angular indurated 75 to 250-millimeter Limestone fragments and 5 percent nonflat angular indurated 2 to 75-millimeter Limestone fragments; slight effervescence, by HCl, 1 normal; clear wavy boundary.

Bw--15 to 31 centimeters (5.9 to 12.2 inches); extremely gravelly clay, very dark gray (10YR 3/1), moist; moderate fine granular structure; slightly hard, friable; common very fine roots throughout and common medium roots throughout and common fine roots throughout and few coarse roots throughout; 25 percent flat subangular 2 to 150-millimeter Petrocalcic fragments and 40 percent nonflat subangular indurated 2 to 75-millimeter Limestone fragments; strong effervescence, by HCl, 1 normal; abrupt wavy boundary.

Bkkm--31 to 50 centimeters (12.2 to 19.7 inches); cemented material; few very fine roots in cracks and few medium roots in cracks and few fine roots in cracks and few coarse roots in cracks; soil in cracks, ; abrupt wavy boundary.

Cr--50 centimeters (19.7 inches); bedrock; .

**PEDON DESCRIPTION (Trench 2)**

**Print Date:** Jun 11 2019
**Description Date:** Mar 13 2019
**Describer:** Ashley Anderson, Travis Waiser, Geraldine Vega
**Site ID:** S2019TX1370009

**Pedon ID:** S2019TX1370009

**Site Note:**

**Pit Location:**
**Pedon Note:**

**Lab Source ID:**
**Lab Pedon #:**

**User Transect ID:**
**Soil Name as Described/Sampled:** Tarrant
**Classification:** Clayey-skeletal, smectitic, thermic Lithic Calciustolls

**Soil Name as Correlated:**

**Classification:**
**Pedon Type:** correlates to named soil
**Pedon Purpose:** research site

**Taxon Kind:** series

**Associated Soils:**
**Physiographic Division:**
**Physiographic Province:**
**Physiographic Section:**

**State Physiographic Area:**

**Local Physiographic Area:**
**Geomorphic Setting:** on summit of interfluve of ridge on dissected plateau
**Upslope Shape:** linear
**Cross Slope Shape:** linear

**Country:**
**State:** Texas
**County:** Edwards
**MLRA:** 81B -- Edwards Plateau, Central Part
**Soil Survey Area:** TX607 -- Edwards and Real Counties, Texas
**Soil Survey Area:** TX607 -- Edwards and Real Counties, Texas

**Map Unit:**
**Quad Name:** Dunbar Draw SE, Texas
**Std Latitude:** 30.2895833
**Std Longitude:** -100.5534500

**Latitude:** 30 degrees 17 minutes 22.50 seconds north
**Longitude:** 100 degrees 33 minutes 12.42 seconds west
**Datum:** WGS84
**UTM Zone:** 14
**UTM Easting:** 350601 meters
**UTM Northing:** 3351896 meters

**Primary Earth Cover:**
**Secondary Earth Cover:**
**Existing Vegetation:** algerita, live oak, redberry juniper, Texas persimmon, Texas pricklypear, Texas wintergrass
**Parent Material:** residuum weathered from limestone
**Bedrock Kind:**
**Bedrock Depth:**
**Bedrock Hardness:**
**Bedrock Fracture Interval:**

**Particle Size Control Section:** 25 to 41 cm.

**Surface Fragments:** 5.0 percent flat indurated 2- to 150-millimeter Limestone fragments

**Description origin:** NASIS

**Description database:** MLRA09_Temple

**Diagnostic Features:** mollic epipedon 0 to 41 cm.
calcic horizon 17 to 41 cm.
lithic contact 41 to cm.

| Top Depth (cm) | Bottom Depth (cm) | Restriction Kind | Restriction Hardness |
|---|---|---|---|
| 41 | | bedrock, lithic | Indurated |

**Cont. Site ID:** S2019TX1370009

**Pedon ID:**
S2019TX1370009

| Slope (%) | Elevation (meters) | Aspect (deg) | MAAT (C) | MSAT (C) | MWAT (C) | MAP (mm) | Frost-Free Days | Drainage Class | Slope Length (meters) | Upslope Length (meters) |
|---|---|---|---|---|---|---|---|---|---|---|
| | 694.9 | | | | | | | | | |

A--0 to 17 centimeters (0.0 to 6.7 inches); silty clay, black (10YR 2/1), moist; moderate fine subangular blocky parts to moderate fine granular structure; hard, friable; common very fine roots throughout and few medium roots throughout and common fine roots throughout and few coarse roots throughout; 5 percent nonflat subangular indurated 2 to 75-millimeter Limestone fragments; slight effervescence, by HCl, 1 normal; clear wavy boundary.

Ak--17 to 41 centimeters (6.7 to 16.1 inches); very dark gray (10YR 3/1) extremely gravelly clay, black (10YR 2/1), moist; moderate fine subangular blocky parts to moderate fine granular structure; hard, friable; common very fine roots throughout and common medium roots throughout and common fine roots throughout and few coarse roots throughout; 5 percent carbonate nodules on bottom of rock fragments; 15 percent flat indurated 2 to 150-millimeter Limestone fragments and 15 percent flat indurated 150 to 350-millimeter Limestone fragments and 35 percent nonflat subangular indurated 2 to 75-millimeter Limestone fragments; strong effervescence, by HCl, 1 normal; abrupt wavy boundary.

R--41 centimeters (16.1 inches); .

**PEDON DESCRIPTION (Trench 4)**

**Print Date:** Jun 11 2019

**Description Date:** Mar 13 2019

**Describer:** Ashley Anderson, Travis Waiser, Geraldine Vega

**Site ID:** P2019TX1370010

**Pedon ID:** P2019TX1370010

**Site Note:**

**Pit Location:**

**Pedon Note:**

**Lab Source ID:**

**Lab Pedon #:**

**User Transect ID:**

**Soil Name as Described/Sampled:** Tarrant

**Classification:** Clayey-skeletal, smectitic, thermic, shallow Typic Calciustolls

**Soil Name as Correlated:**

**Classification:**

**Pedon Type:** taxadjunct to the series

**Pedon Purpose:** research site

**Taxon Kind:** taxadjunct

**Associated Soils:**

**Physiographic Division:**

**Physiographic Province:**

**Physiographic Section:**

**State Physiographic Area:**

**Local Physiographic Area:**

**Geomorphic Setting:** on summit of interfluve of ridge on dissected plateau

**Upslope Shape:** linear

**Cross Slope Shape:** linear

**Country:**

**State:** Texas

**County:** Edwards

**MLRA:** 81B -- Edwards Plateau, Central Part

**Soil Survey Area:** TX607 -- Edwards and Real Counties, Texas

**Soil Survey Area:** TX607 -- Edwards and Real Counties, Texas

**Map Unit:**

**Quad Name:** Dunbar Draw SE, Texas

**Std Latitude:** 30.2799833

**Std Longitude:** -100.5603667

**Latitude:** 30 degrees 16 minutes 47.94 seconds north

**Longitude:** 100 degrees 33 minutes 37.32 seconds west

**Datum:** WGS84

**UTM Zone:** 14

**UTM Easting:** 349921 meters

**UTM Northing:** 3350841 meters

**Primary Earth Cover:**

**Secondary Earth Cover:**

**Existing Vegetation:** live oak, redberry juniper, sacahuista, Texas pricklypear, Texas wintergrass

**Parent Material:** residuum weathered from limestone

**Bedrock Kind:**

**Bedrock Depth:**

**Bedrock Hardness:**

**Bedrock Fracture Interval:**

**Particle Size Control Section:** 25 to 42 cm.

**Surface Fragments:** 40.0 percent nonflat subangular indurated 2- to 75-millimeter Limestone fragments and 5.0 percent nonflat subangular indurated 75- to 250-millimeter Limestone fragments

**Description origin:** NASIS

**Description database:** MLRA09_Temple

**Diagnostic Features:** mollic epidedon 0 to 42 cm.
calcic horizon 16 to 42 cm.
paralithic contact 42 to cm.

| Top Depth (cm) | Bottom Depth (cm) | Restriction Kind | Restriction Hardness |
|---|---|---|---|
| 42 | | bedrock, paralithic | Moderately cemented |

**Cont. Site ID:** P2019TX1370010

**Pedon ID:**
P2019TX1370010

| Slope (%) | Elevation (meters) | Aspect (deg) | MAAT (C) | MSAT (C) | MWAT (C) | MAP (mm) | Frost-Free Days | Drainage Class | Slope Length (meters) | Upslope Length (meters) |
|---|---|---|---|---|---|---|---|---|---|---|
| | 696.8 | | | | | | | | | |

A--0 to 16 centimeters (0.0 to 6.3 inches); clay, black (10YR 2/1), moist; moderate fine granular structure; common very fine roots throughout and common medium roots throughout and common fine roots throughout and few coarse roots throughout; 5 percent nonflat subangular moderately cemented 2 to 75-millimeter Petrocalcic fragments.

Ak--16 to 42 centimeters (6.3 to 16.5 inches); extremely gravelly clay, very dark grayish brown (10YR 3/2), moist; moderate fine granular structure; common very fine roots throughout and few very coarse roots throughout and common medium roots throughout and common fine roots throughout and few coarse roots throughout; 5 percent carbonate nodules on bottom of rock fragments; 20 percent flat moderately cemented 2 to 150-millimeter Petrocalcic fragments and 20 percent flat moderately cemented 150 to 380-millimeter Petrocalcic fragments and 25 percent nonflat subangular moderately cemented 2 to 75-millimeter Petrocalcic fragments.

Cr--42 centimeters (16.5 inches); bedrock; .

**PEDON DESCRIPTION (Trench 5A, location 10 m)**

**Print Date:** Jun 11 2019

**Description Date:** Mar 14 2019

**Describer:** Ashley Anderson, Travis Waiser, Geraldine Vega

**Site ID:** S2019TX1370013

**Pedon ID:** S2019TX1370013

**Site Note:**

**Pit Location:**

**Pedon Note:**

**Lab Source ID:**

**Lab Pedon #:**

**User Transect ID:**

**Soil Name as Described/Sampled:** Ozona

**Classification:** Loamy, mixed, superactive, thermic, shallow Petrocalcic Calciustolls

**Soil Name as Correlated:**

**Classification:**

**Pedon Type:** correlates to named soil

**Pedon Purpose:** research site

**Taxon Kind:** series

**Associated Soils:**

**Physiographic Division:**

**Physiographic Province:**

**Physiographic Section:**

**State Physiographic Area:**

**Local Physiographic Area:**

**Geomorphic Setting:** on backslope of side slope of ridge on dissected plateau

**Upslope Shape:** linear

**Cross Slope Shape:** convex

**Country:**

**State:** Texas

**County:** Edwards

**MLRA:** 81B -- Edwards Plateau, Central Part

**Soil Survey Area:** TX607 -- Edwards and Real Counties, Texas

**Soil Survey Area:** TX607 -- Edwards and Real Counties, Texas

**Map Unit:**

**Quad Name:** Dunbar Draw SE, Texas

**Std Latitude:** 30.2553056

**Std Longitude:** -100.5723333

**Latitude:** 30 degrees 15 minutes 19.10 seconds north

**Longitude:** 100 degrees 34 minutes 20.40 seconds west

**Datum:** WGS84

**UTM Zone:** 14

**UTM Easting:** 348732 meters

**UTM Northing:** 3348122 meters

**Primary Earth Cover:**

**Secondary Earth Cover:**

**Existing Vegetation:** cedar sedge, purple threeawn, redberry juniper

**Parent Material:** residuum weathered from limestone

**Bedrock Kind:**

**Bedrock Depth:**

**Bedrock Hardness:**

**Bedrock Fracture Interval:**

**Particle Size Control Section:** 0 to 34 cm.

**Surface Fragments:** 5.0 percent nonflat subangular strongly cemented 2- to 75-millimeter Limestone fragments and 5.0 percent nonflat subangular strongly cemented 75- to 250-millimeter Limestone fragments

**Description origin:** NASIS

**Description database:** MLRA09_Temple

**Diagnostic Features:** mollic epipedon 0 to 34 cm.
                  petrocalcic horizon 34 to 37 cm.
                  paralithic contact 37 to 200 cm.

| Top Depth (cm) | Bottom Depth (cm) | Restriction Kind | Restriction Hardness |
|---|---|---|---|
| 34 | 37 | petrocalcic | Moderately cemented |
| 37 | 200 | bedrock, paralithic | Weakly cemented |

**Cont. Site ID:** S2019TX1370013

**Pedon ID:**
S2019TX1370013

| Slope (%) | Elevation (meters) | Aspect (deg) | MAAT (C) | MSAT (C) | MWAT (C) | MAP (mm) | Frost-Free Days | Drainage Class | Slope Length (meters) | Upslope Length (meters) |
|---|---|---|---|---|---|---|---|---|---|---|
| | 685.5 | | | | | | | | | |

A1--0 to 20 centimeters (0.0 to 7.9 inches); very dark grayish brown (10YR 3/2) silty clay loam, very dark brown (10YR 2/2), moist; moderate medium granular structure; slightly hard, friable; common very fine roots throughout and few medium roots throughout and common fine roots throughout; 8 percent nonflat subangular indurated 2 to 75-millimeter Limestone fragments; violent effervescence, by HCl, 1 normal; clear wavy boundary.

A2--20 to 34 centimeters (7.9 to 13.4 inches); dark grayish brown (10YR 4/2) extremely channery silty clay loam, very dark brown (10YR 2/2), moist; moderate medium granular structure; slightly hard, friable; common very fine roots throughout and common medium roots throughout and common fine roots throughout and common coarse roots throughout; 30 percent nonflat subangular indurated 2 to 75-millimeter Limestone fragments and 35 percent flat angular indurated 2 to 150-millimeter Limestone fragments; violent effervescence, by HCl, 1 normal; clear wavy boundary.

Bkkm--34 to 37 centimeters (13.4 to 14.6 inches); material; clear wavy boundary.

Cr--37 to 200 centimeters (14.6 to 78.7 inches); bedrock; very few very fine roots throughout and few medium roots throughout and very few fine roots throughout; Small pocket at 100 to 130 cm of soil material in Cr was silty clay with 17% sand, 42% silt, and 41% clay.

**PEDON DESCRIPTION (Trench 5A, location 18 m)**

**Print Date:** Jun 11 2019
**Description Date:** Mar 14 2019
**Describer:** Ashley Anderson, Travis Waiser, Geraldine Vega
**Site ID:** P2019TX1370014

**Pedon ID:** P2019TX1370014

**Site Note:**

**Pit Location:**
**Pedon Note:**

**Lab Source ID:**
**Lab Pedon #:**

**User Transect ID:**
**Soil Name as Described/Sampled:** Prade
**Classification:** Clayey-skeletal, smectitic, thermic, shallow Petrocalcic Calciustolls

**Soil Name as Correlated:**

**Classification:**
**Pedon Type:** correlates to named soil
**Pedon Purpose:** research site

**Taxon Kind:** series

**Associated Soils:**
**Physiographic Division:**
**Physiographic Province:**
**Physiographic Section:**

**State Physiographic Area:**

**Local Physiographic Area:**
**Geomorphic Setting:** on backslope of side slope of ridge on dissected plateau
**Upslope Shape:** linear
**Cross Slope Shape:** convex

**Country:**
**State:** Texas
**County:** Edwards
**MLRA:** 81B -- Edwards Plateau, Central Part
**Soil Survey Area:** TX607 -- Edwards and Real Counties, Texas
**Soil Survey Area:** TX607 -- Edwards and Real Counties, Texas

**Map Unit:**
**Quad Name:** Dunbar Draw SE, Texas
**Std Latitude:** 30.2555833
**Std Longitude:** -100.5726389

**Latitude:** 30 degrees 15 minutes 20.10 seconds north
**Longitude:** 100 degrees 34 minutes 21.50 seconds west
**Datum:** WGS84
**UTM Zone:** 14
**UTM Easting:** 348703 meters
**UTM Northing:** 3348153 meters

**Primary Earth Cover:**
**Secondary Earth Cover:**
**Existing Vegetation:** cedar sedge, purple threeawn, redberry juniper
**Parent Material:** residuum weathered from limestone
**Bedrock Kind:**
**Bedrock Depth:**
**Bedrock Hardness:**
**Bedrock Fracture Interval:**

**Particle Size Control Section:** 25 to 41 cm.

**Surface Fragments:** 5.0 percent nonflat subangular strongly cemented 2- to 75-millimeter Limestone fragments and 5.0 percent nonflat subangular strongly cemented 75- to 250-millimeter Limestone fragments

**Description origin:** NASIS

**Description database:** MLRA09_Temple

**Diagnostic Features:** mollic epipedon 0 to 41 cm.
petrocalcic horizon 41 to 43 cm.
paralithic materials 43 to 86 cm.

| Top Depth (cm) | Bottom Depth (cm) | Restriction Kind | Restriction Hardness |
|---|---|---|---|
| 41 | 43 | petrocalcic | Strongly cemented |
| 43 | 86 | bedrock, paralithic | Moderately cemented |

**Cont. Site ID:** P2019TX1370014

**Pedon ID:**
P2019TX1370014

| Slope (%) | Elevation (meters) | Aspect (deg) | MAAT (C) | MSAT (C) | MWAT (C) | MAP (mm) | Frost-Free Days | Drainage Class | Slope Length (meters) | Upslope Length (meters) |
|---|---|---|---|---|---|---|---|---|---|---|
| | 691.0 | | | | | | | | | |

A1--0 to 21 centimeters (0.0 to 8.3 inches); very dark grayish brown (10YR 3/2) clay, very dark brown (10YR 2/2), moist; moderate medium granular structure; slightly hard, friable; common very fine roots throughout and few medium roots throughout and common fine roots throughout; 6 percent nonflat subangular indurated 2 to 75-millimeter Limestone fragments; violent effervescence, by HCl, 1 normal; clear wavy boundary.

A2--21 to 41 centimeters (8.3 to 16.1 inches); dark grayish brown (10YR 4/2) extremely gravelly clay, very dark brown (10YR 2/2), moist; moderate medium granular structure; slightly hard, friable; common very fine roots throughout and few very coarse roots throughout and few medium roots throughout and common fine roots throughout and few coarse roots throughout; 10 percent flat angular moderately cemented 2 to 150-millimeter Petrocalcic fragments and 25 percent flat angular moderately cemented 150 to 350-millimeter Petrocalcic fragments and 30 percent nonflat subangular indurated 2 to 75-millimeter Limestone fragments; violent effervescence, by HCl, 1 normal; clear wavy boundary.

Bkkm--41 to 43 centimeters (16.1 to 16.9 inches); material; clear wavy boundary.

Cr--43 to 86 centimeters (16.9 to 33.9 inches); bedrock; .

**PEDON DESCRIPTION (Trench 5B)**

**Print Date:** Jun 11 2019
**Description Date:** Mar 14 2019
**Describer:** Ashley Anderson, Travis Waiser, Geraldine Vega
**Site ID:** S2019TX1370012

**Pedon ID:** S2019TX1370012

**Site Note:**

**Pit Location:**
**Pedon Note:**

**Lab Source ID:**
**Lab Pedon #:**

**User Transect ID:**
**Soil Name as Described/Sampled:** Rio Diablo
**Classification:** Fine, mixed, superactive, thermic Pachic Haplustolls

**Soil Name as Correlated:**

**Classification:**
**Pedon Type:** taxadjunct to the series
**Pedon Purpose:** research site

**Taxon Kind:** taxadjunct

**Associated Soils:**
**Physiographic Division:**
**Physiographic Province:**
**Physiographic Section:**

**State Physiographic Area:**

**Local Physiographic Area:**
**Geomorphic Setting:** on footslope of base slope of ridge on dissected plateau
**Upslope Shape:** concave
**Cross Slope Shape:** linear

**Country:**
**State:** Texas
**County:** Edwards
**MLRA:** 81B -- Edwards Plateau, Central Part
**Soil Survey Area:** TX607 -- Edwards and Real Counties, Texas
**Soil Survey Area:** TX607 -- Edwards and Real Counties, Texas

**Map Unit:**
**Quad Name:** Dunbar Draw SE, Texas
**Std Latitude:** 30.2552500
**Std Longitude:** -100.5721667

**Latitude:** 30 degrees 15 minutes 18.90 seconds north
**Longitude:** 100 degrees 34 minutes 19.80 seconds west
**Datum:** WGS84
**UTM Zone:** 14
**UTM Easting:** 348748 meters
**UTM Northing:** 3348115 meters

**Primary Earth Cover:**
**Secondary Earth Cover:**
**Existing Vegetation:** Christmas cactus, honey mesquite, redberry juniper, Texas pricklypear
**Parent Material:** alluvium derived from limestone
**Bedrock Kind:**

**Bedrock Depth:**

**Bedrock Hardness:**
**Bedrock Fracture Interval:**

**Particle Size Control Section:** 25 to 100 cm.

**Surface Fragments:** 2.0 percent nonflat subrounded indurated 2- to 75-millimeter Limestone fragments

**Description origin:** NASIS

**Description database:** MLRA09_Temple

**Diagnostic Features:** mollic epipedon 0 to 60 cm.
cambic horizon 60 to 118 cm.
paralithic materials 118 to 147 cm.

| Top Depth (cm) | Bottom Depth (cm) | Restriction Kind | Restriction Hardness |
|---|---|---|---|
| 118 | 147 | bedrock, paralithic | Moderately cemented |

**Cont. Site ID:** S2019TX1370012

**Pedon ID:** S2019TX1370012

| Slope (%) | Elevation (meters) | Aspect (deg) | MAAT (C) | MSAT (C) | MWAT (C) | MAP (mm) | Frost-Free Days | Drainage Class | Slope Length (meters) | Upslope Length (meters) |
|---|---|---|---|---|---|---|---|---|---|---|
| | 691.0 | | | | | | | | | |

A1--0 to 18 centimeters (0.0 to 7.1 inches); very dark grayish brown (10YR 3/2) silty clay loam, very dark brown (10YR 2/2), moist; weak medium subangular blocky parts to moderate fine granular structure; slightly hard, friable; common very fine roots throughout and common fine roots throughout; 2 percent nonflat subangular indurated 2 to 20-millimeter Limestone fragments; violent effervescence, by HCl, 1 normal; clear smooth boundary.

A2--18 to 35 centimeters (7.1 to 13.8 inches); very dark grayish brown (10YR 3/2) silty clay, very dark brown (10YR 2/2), moist; weak medium subangular blocky parts to moderate fine granular structure; slightly hard, friable; common very fine roots throughout and few medium roots throughout and common fine roots throughout; 4 percent nonflat subangular indurated 2 to 20-millimeter Limestone fragments; violent effervescence, by HCl, 1 normal; clear smooth boundary.

Bw--35 to 60 centimeters (13.8 to 23.6 inches); brown (7.5YR 4/3) silty clay, dark brown (10YR 3/3), moist; moderate medium subangular blocky structure; hard, firm; common very fine roots throughout and few medium roots throughout and common fine roots throughout and few coarse roots throughout; common very fine tubular pores; 6 percent nonflat subangular indurated 2 to 75-millimeter Limestone fragments; violent effervescence, by HCl, 1 normal; gradual wavy boundary.

Bk1--60 to 92 centimeters (23.6 to 36.2 inches); brown (7.5YR 4/4) silty clay, brown (7.5YR 4/4), moist; weak medium subangular blocky parts to moderate fine subangular blocky structure; hard, firm; common very fine roots throughout and few medium roots throughout and common fine roots throughout and few coarse roots throughout; common very fine tubular pores; 2 percent fine threadlike carbonate masses throughout; 8 percent nonflat subangular indurated 2 to 75-millimeter Limestone fragments; violent

effervescence, by HCl, 1 normal; gradual wavy boundary.

Bk2--92 to 118 centimeters (36.2 to 46.5 inches); strong brown (7.5YR 5/6) silty clay, strong brown (7.5YR 5/6), moist; moderate fine subangular blocky structure; hard, firm; few very fine roots throughout and few medium roots throughout and few fine roots throughout; common very fine tubular and common fine tubular pores; 4 percent fine threadlike carbonate masses throughout; 8 percent nonflat subangular indurated 2 to 75-millimeter Limestone fragments; violent effervescence, by HCl, 1 normal.

Cr--118 to 147 centimeters (46.5 to 57.9 inches); bedrock; few fine roots throughout; violent effervescence, by HCl, 1 normal.

**PEDON DESCRIPTION (Trench 6, location 4 m)**

**Print Date:** Jun 11 2019
**Description Date:** Mar 12 2019
**Describer:** Ashley Anderson, Travis Waiser, Geraldine Vega
**Site ID:** P2019TX1370002

**Pedon ID:** P2019TX1370002

**Site Note:**

**Pit Location:**
**Pedon Note:**

**Lab Source ID:**
**Lab Pedon #:**

**User Transect ID:**
**Soil Name as Described/Sampled:** Mereta
**Classification:** Clayey, mixed, superactive, thermic, shallow Petrocalcic Calciustolls

**Soil Name as Correlated:**

**Classification:**
**Pedon Type:** correlates to named soil
**Pedon Purpose:** research site

**Taxon Kind:** series

**Associated Soils:**
**Physiographic Division:**
**Physiographic Province:**
**Physiographic Section:**

**State Physiographic Area:**

**Local Physiographic Area:**
**Geomorphic Setting:** on toeslope of base slope of ridge on dissected plateau
**Upslope Shape:** linear
**Cross Slope Shape:** linear

**Country:**
**State:** Texas
**County:** Edwards
**MLRA:** 81B -- Edwards Plateau, Central Part
**Soil Survey Area:** TX607 -- Edwards and Real Counties, Texas
**Soil Survey Area:** TX607 -- Edwards and Real Counties, Texas

**Map Unit:**
**Quad Name:** Dunbar Draw SE, Texas
**Std Latitude:** 30.2833000
**Std Longitude:** -100.5410667

**Latitude:** 30 degrees 16 minutes 59.88 seconds north
**Longitude:** 100 degrees 32 minutes 27.84 seconds west
**Datum:** WGS84
**UTM Zone:** 14
**UTM Easting:** 351783 meters
**UTM Northing:** 3351183 meters

**Primary Earth Cover:**
**Secondary Earth Cover:**
**Existing Vegetation:** honey mesquite, live oak, redberry juniper, Texas pricklypear, Texas wintergrass
**Parent Material:** alluvium derived from limestone
**Bedrock Kind:**
**Bedrock Depth:**
**Bedrock Hardness:**
**Bedrock Fracture Interval:**

**Particle Size Control Section:** 25 to 39 cm.

**Surface Fragments:** 5.0 percent nonflat subangular indurated 2- to 75-millimeter Limestone fragments and 5.0 percent nonflat subangular indurated 75- to 250-millimeter Limestone fragments

**Description origin:** NASIS

**Description database:** MLRA09_Temple

**Diagnostic Features:** mollic epipedon 0 to 39 cm.
petrocalcic horizon 39 to 65 cm.
cambic horizon 65 to 145 cm.
paralithic contact 145 to 155 cm.

| Top Depth (cm) | Bottom Depth (cm) | Restriction Kind | Restriction Hardness |
|---|---|---|---|
| 39 | 65 | petrocalcic | Strongly cemented |
| 145 | 155 | bedrock, paralithic | Weakly cemented |

**Cont. Site ID:** P2019TX1370002

**Pedon ID:** P2019TX1370002

| Slope (%) | Elevation (meters) | Aspect (deg) | MAAT (C) | MSAT (C) | MWAT (C) | MAP (mm) | Frost-Free Days | Drainage Class | Slope Length (meters) | Upslope Length (meters) |
|---|---|---|---|---|---|---|---|---|---|---|
| | 644.3 | | | | | | | | | |

A1--0 to 20 centimeters (0.0 to 7.9 inches); very dark gray (10YR 3/1) clay, very dark brown (10YR 2/2), moist; moderate medium subangular blocky, and moderate fine subangular blocky structure; slightly hard, friable; common very fine roots throughout and few medium roots throughout and common fine roots throughout; 8 percent nonflat subrounded indurated 2 to 75-millimeter Limestone fragments; violent effervescence, by HCl, 1 normal; clear smooth boundary.

A2--20 to 39 centimeters (7.9 to 15.4 inches); dark grayish brown (10YR 4/2) gravelly clay, dark brown (10YR 3/3), moist; moderate fine subangular blocky structure; hard, firm; common very fine roots throughout and few very coarse roots throughout and common medium roots throughout and common fine roots throughout and few coarse roots throughout; 1 percent nonflat subrounded indurated 75 to 250-millimeter Limestone fragments and 15 percent nonflat subangular indurated 2 to 75-millimeter Limestone fragments; violent effervescence, by HCl, 1 normal; clear wavy boundary.

Bkkm--39 to 65 centimeters (15.4 to 25.6 inches); cemented material; few very fine roots throughout and few medium roots throughout and few fine roots throughout; violent effervescence, by HCl, 1 normal; abrupt wavy boundary.

Bk1--65 to 107 centimeters (25.6 to 42.1 inches); brown (7.5YR 4/3) clay, brown (7.5YR 4/3), moist; weak fine subangular blocky structure; slightly hard, friable; common very fine roots throughout and few very coarse roots throughout and common fine roots throughout and few coarse roots throughout; 3 percent fine threadlike carbonate masses; 2 percent nonflat subrounded indurated 75 to 250-millimeter Limestone fragments and 10 percent nonflat subrounded indurated 2 to 75-millimeter Limestone fragments; violent effervescence, by HCl, 1 normal; clear wavy boundary.

Bk2--107 to 145 centimeters (42.1 to 57.1 inches); light brown (7.5YR 6/4) clay, brown (7.5YR 5/4), moist; weak fine subangular blocky structure; slightly hard, friable; common very fine roots throughout and few medium roots throughout and common fine roots throughout and few coarse roots throughout; 3 percent fine threadlike carbonate masses; 1 percent nonflat subrounded indurated 75 to 250-millimeter Limestone fragments and 8 percent nonflat subrounded indurated 2 to 75-millimeter Limestone fragments; violent effervescence, by HCl, 1 normal; abrupt wavy boundary.

Cr--145 to 155 centimeters (57.1 to 61.0 inches); bedrock; violent effervescence, by HCl, 1 normal.

**PEDON DESCRIPTION (Trench 6, location 6 m)**

**Print Date:** Jun 11 2019

**Description Date:** Mar 12 2019

**Describer:** Ashley Anderson, Travis Waiser, Geraldine Vega

**Site ID:** S2019TX1370001

**Pedon ID:** S2019TX1370001

**Site Note:**

**Pit Location:**

**Pedon Note:**

**Lab Source ID:**

**Lab Pedon #:**

**User Transect ID:**

**Soil Name as Described/Sampled:** Mereta

**Classification:** Clayey, mixed, superactive, thermic, shallow Petrocalcic Calciustolls

**Soil Name as Correlated:**

**Classification:**

**Pedon Type:** correlates to named soil

**Pedon Purpose:** research site

**Taxon Kind:** series

**Associated Soils:**

**Physiographic Division:**

**Physiographic Province:**

**Physiographic Section:**

**State Physiographic Area:**

**Local Physiographic Area:**

**Geomorphic Setting:** on toeslope of base slope of ridge on dissected plateau

**Upslope Shape:** linear

**Country:**

**State:** Texas

**County:** Edwards

**MLRA:** 81B -- Edwards Plateau, Central Part

**Soil Survey Area:** TX607 -- Edwards and Real Counties, Texas

**Soil Survey Area:** TX607 -- Edwards and Real Counties, Texas

**Map Unit:**

**Quad Name:** Dunbar Draw SE, Texas

**Std Latitude:** 30.2833833

**Std Longitude:** -100.5411167

**Latitude:** 30 degrees 17 minutes 0.18 seconds north

**Longitude:** 100 degrees 32 minutes 28.02 seconds west

**Datum:** WGS84

**UTM Zone:** 14

**UTM Easting:** 351778 meters

**UTM Northing:** 3351193 meters

**Primary Earth Cover:** Grass/herbaceous cover

**Secondary Earth Cover:** Savanna rangeland

**Existing Vegetation:** honey mesquite, live oak, redberry juniper, Texas pricklypear, Texas wintergrass

**Parent Material:** alluvium derived from limestone

**Bedrock Kind:**

**Bedrock Depth:**

**Bedrock Hardness:**

**Cross Slope Shape:** linear

**Particle Size Control Section:** 25 to 43 cm.

**Description origin:** NASIS

**Diagnostic Features:** mollic epipedon 0 to 43 cm.
                                 petrocalcic horizon 43 to 55 cm.

**Bedrock Fracture Interval:**

**Surface Fragments:** 5.0 percent nonflat subangular indurated 2- to 75-millimeter Limestone fragments and 5.0 percent nonflat subangular indurated 75- to 250-millimeter Limestone fragments

**Description database:** MLRA09_Temple

| Top Depth (cm) | Bottom Depth (cm) | Restriction Kind | Restriction Hardness |
|---|---|---|---|
| 43 | 55 | petrocalcic | Weakly cemented |

**Cont. Site ID:** S2019TX1370001

**Pedon ID:**
S2019TX1370001

| Slope (%) | Elevation (meters) | Aspect (deg) | MAAT (C) | MSAT (C) | MWAT (C) | MAP (mm) | Frost-Free Days | Drainage Class | Slope Length (meters) | Upslope Length (meters) |
|---|---|---|---|---|---|---|---|---|---|---|
| | 638.6 | | | | | | | | | |

A1--0 to 23 centimeters (0.0 to 9.1 inches); very dark grayish brown (10YR 3/2) clay, very dark brown (10YR 2/2), moist; moderate medium subangular blocky, and moderate fine subangular blocky structure; slightly hard, friable; common very fine roots throughout and common medium roots throughout and common fine roots throughout and few coarse roots throughout; few fine tubular pores; 8 percent nonflat subrounded indurated 2 to 75-millimeter Limestone fragments; violent effervescence, by HCl, 1 normal; gradual smooth boundary.

A2--23 to 43 centimeters (9.1 to 16.9 inches); dark grayish brown (10YR 4/2) clay, dark brown (10YR 3/3), moist; moderate medium subangular blocky structure; hard, firm; common very fine roots throughout and very few very coarse roots throughout and common medium roots throughout and few fine roots throughout and few coarse roots throughout; 1 percent nonflat subangular indurated 75 to 255-millimeter Limestone fragments and 8 percent nonflat subrounded indurated 2 to 75-millimeter Limestone fragments; violent effervescence, by HCl, 1 normal; abrupt wavy boundary.

Bkkm--43 to 55 centimeters (16.9 to 21.7 inches); cemented material; few very fine roots in cracks and few fine roots in cracks; violent effervescence, by HCl, 1 normal; gradual wavy boundary.

Ck1--55 to 125 centimeters (21.7 to 49.2 inches); pink (7.5YR 8/3) material; few very fine roots in cracks

and few medium roots in cracks and few fine roots in cracks; violent effervescence, by HCl, 1 normal; gradual wavy boundary.

Ck2--125 to 180 centimeters (49.2 to 70.9 inches); pink (7.5YR 8/3) material; few very fine roots in cracks and few medium roots in cracks and few fine roots in cracks; 10 percent coarse carbonate masses and 10 percent coarse carbonate nodules; violent effervescence, by HCl, 1 normal.

**PEDON DESCRIPTION (Trench 6, location 8 m)**

**Print Date:** Jun 11 2019
**Description Date:** Mar 12 2019
**Describer:** Ashley Anderson, Travis Waiser, Geraldine Vega
**Site ID:** S2019TX1370003

**Pedon ID:** S2019TX1370003

**Site Note:**

**Pit Location:**
**Pedon Note:**

**Lab Source ID:**
**Lab Pedon #:**

**User Transect ID:**
**Soil Name as Described/Sampled:** Rio Diablo
**Classification:** Fine, mixed, superactive, thermic Aridic Haplustolls

**Soil Name as Correlated:**

**Classification:**
**Pedon Type:** correlates to named soil
**Pedon Purpose:** research site

**Taxon Kind:** series

**Associated Soils:**
**Physiographic Division:**
**Physiographic Province:**
**Physiographic Section:**

**State Physiographic Area:**

**Local Physiographic Area:**
**Geomorphic Setting:** on toeslope of base slope of ridge on dissected plateau
**Upslope Shape:** linear
**Cross Slope Shape:** linear

**Country:**
**State:** Texas
**County:** Edwards
**MLRA:** 81B -- Edwards Plateau, Central Part
**Soil Survey Area:** TX607 -- Edwards and Real Counties, Texas
**Soil Survey Area:** TX607 -- Edwards and Real Counties, Texas

**Map Unit:**
**Quad Name:** Dunbar Draw SE, Texas
**Std Latitude:** 30.2834000
**Std Longitude:** -100.5411667

**Latitude:** 30 degrees 17 minutes 0.24 seconds north
**Longitude:** 100 degrees 32 minutes 28.20 seconds west
**Datum:** WGS84
**UTM Zone:** 14
**UTM Easting:** 351773 meters
**UTM Northing:** 3351195 meters

**Primary Earth Cover:**
**Secondary Earth Cover:**
**Existing Vegetation:** curly-mesquite, honey mesquite, redberry juniper, Texas pricklypear, Texas wintergrass
**Parent Material:** alluvium derived from limestone
**Bedrock Kind:**
**Bedrock Depth:**
**Bedrock Hardness:**
**Bedrock Fracture Interval:**

**Particle Size Control Section:** 25 to 100 cm.

**Surface Fragments:** 5.0 percent nonflat subangular indurated 2- to 75-millimeter Limestone fragments and 5.0 percent nonflat subangular indurated 75- to 250-millimeter Limestone fragments

**Description origin:** NASIS

**Description database:** MLRA09_Temple

**Diagnostic Features:** mollic epipedon 0 to 29 cm.
cambic horizon 29 to 120 cm.

**Cont. Site ID:** S2019TX1370003

**Pedon ID:**
S2019TX1370003

| Slope (%) | Elevation (meters) | Aspect (deg) | MAAT (C) | MSAT (C) | MWAT (C) | MAP (mm) | Frost-Free Days | Drainage Class | Slope Length (meters) | Upslope Length (meters) |
|---|---|---|---|---|---|---|---|---|---|---|
| | 675.4 | | | | | | | | | |

A--0 to 29 centimeters (0.0 to 11.4 inches); very dark gray (10YR 3/1) silty clay, black (10YR 2/1), moist; strong fine subangular blocky parts to moderate fine granular structure; slightly hard, friable; common very fine roots throughout and common fine roots throughout; 4 percent nonflat subrounded indurated 2 to 75-millimeter Limestone fragments; violent effervescence, by HCl, 1 normal; clear smooth boundary.

Bw--29 to 61 centimeters (11.4 to 24.0 inches); brown (7.5YR 4/3) clay, brown (7.5YR 4/3), moist; moderate medium subangular blocky, and moderate medium angular blocky structure; hard, firm; common very fine roots throughout and few medium roots throughout and common fine roots throughout; 8 percent nonflat subrounded indurated 2 to 75-millimeter Limestone fragments; violent effervescence, by HCl, 1 normal; clear smooth boundary.

Bk1--61 to 100 centimeters (24.0 to 39.4 inches); weak red (7.5R 4/3) clay, brown (7.5YR 4/3), moist; moderate medium prismatic structure; hard, firm; common very fine roots throughout and common fine roots throughout; 4 percent fine threadlike carbonate masses; 1 percent nonflat subrounded indurated 75 to 250-millimeter Limestone fragments and 10 percent nonflat subrounded indurated 2 to 75-millimeter Limestone fragments; violent effervescence, by HCl, 1 normal; clear smooth boundary.

Bk2--100 to 120 centimeters (39.4 to 47.2 inches); light brown (7.5YR 6/4) clay, red (7.5R 5/6), moist; weak medium subangular blocky structure; hard, firm; few very fine roots throughout and few fine roots throughout; 3 percent fine spherical carbonate masses; 1 percent nonflat subrounded indurated 75 to 250-millimeter Limestone fragments and 8 percent nonflat subrounded indurated 2 to 75-millimeter Limestone fragments; violent effervescence, by HCl, 1 normal.